# Signaling cascades shape functional subpopulations of cortical astrocytes in male wild-type mice and APP/PS1dE9 Alzheimer's disease model

Yiannis Poulot-Becq-Giraudon [1,2], Océane Guillemaud[2,8], Elisa Degl'Innocenti [1,8], Vivien Letenneur[2,9], Karouna Bascarane[1,2,9], Tony Barbay [3,9], Mie Møller Clausen [2,9], Céline Derbois[4,9], Martine Guillermier[1,2,9], Ludmila Juricek[2], Miriam Riquelme-Perez [2,4], Tom Lakomy [1,2], Lucile Benhaim[1,2], Noëlle Dufour[2], Pauline Gipchtein[2], Fanny Petit[2], Léa Siron[1], Gwennaëlle Aurégan[2], Nathalie Dechamps[5,6], Marie-Claude Gaillard[2], Alexis-Pierre Bemelmans [2], Rémi Bos [3], Maria-Angeles Carrillo-de Sauvage [1,2], Giampaolo Milior [7], Nathalie Rouach [7], Solène Brohard [4,10], Kevin Muret [4,10], Eric Bonnet [4] & Carole Escartin [1,2] ✉

Astrocytes are key partners for neurons and can impact diseases such as Alzheimer's disease (AD), as they exhibit multiple reactive changes. Recent single cell/nucleus genomics analyses evidence astrocyte subpopulations coexisting in normal and AD brains. However, the signaling cascades controlling them, their functional characteristics and roles in AD are still unknown. Here, thanks to astrocyte-specific reporters for STAT3 and NF-kB signaling pathways, two regulators of astrocyte reactivity, we report the presence of three astrocyte subpopulations defined by their signaling activity, in the prefrontal cortex of male APP/PS1dE9 mice. These subpopulations are not triggered by amyloid deposition and are also observed in wild-type mice. They show distinct morphologies, molecular signatures and functional profiles. While NF-kB+ astrocytes have larger territories and higher lysosomal activity, STAT3+ astrocytes display enhanced hemichannel activity. Specific inhibition of these subpopulations reduces amyloid plaque size and impacts anxiety, social preference and social memory in AD but not wild-type mice. Our results show how innate signaling shapes astrocyte subpopulations in the mouse cortex, with distinct functions in health and disease.

Astrocytes are key elements in brain circuits. They promote brain homeostasis and regulate synaptic activity and plasticity[1]. These morphologically complex cells engage in fine and numerous interactions with brain cells, in a region- and circuit-specific manner[2]. Dating back from their first observations, astrocytes were shown to differ both between and within brain regions, in regards to their morphology but also some specific markers expressed at variable levels, including the widely used astrocyte marker Glial Fibrillary Acidic Protein (GFAP)[3].

More recent transcriptomics studies confirmed that astrocytes are heterogeneous between developmental stages and brain regions[4,5]. Transcriptomics studies at the single cell or nucleus level have further shown that even within a single brain region [e.g., hippocampus[6], cortex[7]], astrocytes form identifiable molecular clusters, associated with potential specific functions. However, these putative functions are rarely validated or assessed[8]. To date, because of technical challenges, only a few studies report functional differences between astrocytes within the same brain region: different ensembles of striatal astrocytes respond to the activation of striatal neurons expressing the dopamine D1 or D2 receptor[9], astrocytes from different cortical layers display different calcium dynamics[6,10] and a morphologically distinct astrocyte subtype expressing oxytocin receptors mediates oxytocin anxiolytic effects in the rodent central amygdala[11].

Pathological conditions introduce an additional level of complexity, as astrocytes change and become reactive. For example, in Alzheimer's disease (AD) models or patients, single cell/nucleus RNA-seq analyses show that several astrocyte clusters vary in proportion or in their gene expression profile with disease[12–14]. But again, the functional differences between these astrocyte clusters are only inferred from gene ontology analysis and are not always validated in situ. Moreover, it is unclear how such subpopulations of molecularly distinct astrocytes are generated or controlled. Intriguingly, these astrocyte clusters are already found in control conditions, suggesting that AD does not result in the appearance of new cell clusters, but rather affects the molecular profile or abundance of pre-existing subpopulations.

Several signaling cascades can be activated in astrocytes in pathological situations, driving various changes in their morphology and transcriptome[15]. Among them, the Signal Transduction and Activator of Transcription 3 (STAT3)[16] and Nuclear Factor κB (NF-kB)[17] pathways emerge as master regulators of reactive astrocytes. Both pathways were found activated in astrocytes in AD contexts, regulating key aspects of reactive astrocyte phenotype[18–23]. Interestingly, these cascades are also important during early brain development to regulate different astrogliogenesis stages[24,25]. Both pathways are activated by a range of mediators such as cytokines, growth factors, and other damage- or pathogen-associated molecular patterns. Phosphorylation cascades initiated at membrane-bound receptors result in the nuclear accumulation of activated STAT3 and NF-kB transcription factors, which subsequently activate the transcription of specific target genes. Both pathways are subject to a tight inhibitory retro-control by phosphatases and specific inhibitors such as Suppressor of Cytokine Signaling 3 (SOCS3) and NF-kB inhibitor A (NFKBIA, also called IkB), respectively[26,27].

In this work, we hypothesized that the STAT3 and NF-kB pathways, as key astrocyte signaling pathways, are differentially activated in AD astrocytes and control their molecular signature and functional features, thereby generating distinct reactive astrocyte subtypes. The active, phosphorylated forms of STAT3 and NF-kB are unreliably detected in the mouse brain, especially in chronic diseases with mild and progressive activation. Therefore, to assess the activity of these two pathways in situ and understand their role in generating reactive astrocyte heterogeneity, we developed two lentivirus-based reporters. These fluorescent reporters reveal distinct astrocyte subpopulations, differing by their activity of STAT3 and NF-kB signaling cascades in the mouse prefrontal cortex (PFC). Thanks to these fluorescent reporters, we could further define the morphology, molecular signature of these subpopulations, and also probe their functional features and reveal their specific roles in a β-amyloidosis mouse model.

## Results

### Tracking astrocyte signaling cascades in situ

To efficiently monitor the activity of two central signaling cascades regulating astrocyte changes in AD, we generated two lentivirus (LV)-based reporters, circumventing the low detection of phosphorylated active forms of STAT3 and NF-kB by immunostaining. They contain a synthetic promoter with six STAT3 or NF-kB consensus responsive elements upstream of a minimal promoter, which controls the expression of cytoplasmic GFP or nuclear eCerulean (CFPnuc), respectively (Fig. 1a). These LV reporters (later called LV-P$_{STAT3}$-GFP and LV-P$_{NF-kB}$-CFPnuc) were pseudotyped with the Mokola envelope to target astrocytes[28]. They were co-injected in the PFC of wild-type (WT) and APP/PS1dE9 (APP/PS1) mice at 10-month-old, a stage with significant amyloid deposition, functional alterations and behavioral deficits (Fig. 1a)[29]. The PFC was selected as a vulnerable brain region in AD[30,31] displaying a high transduction efficiency with Mokola-pseudotyped LV. Two months post-injection, we confirmed the astrocyte-specific targeting of LV reporters, using immunofluorescent stainings with cell-type specific markers. Cells expressing GFP and/or CFPnuc had a typical astrocyte morphology and co-expressed the astrocyte protein phosphoglycerate dehydrogenase (PHGDH)[32], but not the microglial marker Iba1, the neuronal marker NeuN, the oligodendrocyte progenitor cell marker PDGFRα, or the oligodendrocyte marker MOG (Fig. 1b, c). As shown in Fig. 1b, the injection site shows the expected neuronal density and microglia phenotype with a normal, ramified morphology, suggesting minimal injection-induced inflammation or toxicity.

We then sought to validate the functionality of these reporters by controlling that they were sensitive to known pathway inhibitors. First, we used pharmacological inhibitors in vitro. Stattic and BAY 11-7082 (STAT3 and NF-kB pathway selective inhibitors respectively), were applied to HEK cells previously infected with LV-P$_{STAT3}$-GFP or LV-P$_{NF-kB}$-CFPnuc and 10 ng/ml IL6 or TNFα to stimulate each pathway. Both inhibitors significantly reduced reporter fluorescence (Fig. 1d, e). We then performed an in vivo experiment using genetic pathway inhibitors. We co-injected a LV reporter with a LV encoding the endogenous inhibitor: SOCS3 for the JAK-STAT3 pathway and NFKBIA for the NF-kB pathway under STAT3 and NF-kB promoters, respectively (Fig. 1f, g). SOCS3 significantly reduced GFP fluorescence intensity by 26% (Fig. 1f), while NFKBIA reduced CFP fluorescence intensity by more than 50% (Fig. 1g), compared with controls.

Overall, we show that these LV reporters are sensitive to STAT3 and NF-kB activities, specifically in astrocytes of the mouse PFC.

### Active signaling pathways define three astrocyte subpopulations in the mouse PFC

We first focused on the disease context by co-injecting the two LV reporters in the PFC of 12-month-old APP/PS1 mice. We evidenced three subpopulations of fluorescent astrocytes based on their active signaling pathways: GFP+ astrocytes showing STAT3 activity, nuclear CFP+ astrocytes showing NF-kB activity and double fluorescent astrocytes displaying both active cascades (Fig. 2a). Cells of these three subpopulations co-expressed the reactive marker GFAP+, which was induced in these APP/PS1 mice (Fig. 2a), and the astrocyte marker PHGDH (Supplementary Fig. 1a, and Fig. 1b). Fluorescent astrocytes were located randomly over the upper and lower cortical layers, without obvious organization (Fig. 2b). Within the global LV-targeted volume in the PFC, most GFAP+ astrocytes expressed fluorescent reporter proteins and only a rare subpopulation of GFAP+/GFP-/CFP-astrocytes was observed (Fig. 2a). This fourth subpopulation may originate from a lack of transduction by LV reporters, or both cascades being inactive. Since we could not discriminate between the two hypotheses, this subpopulation was not studied further.

We measured the fluorescent levels of GFP and CFP in single astrocyte soma. The resulting dot-plot confirms that each astrocyte displayed a unique combination of GFP and CFP levels ranging from being undetectable to highly expressed (Fig. 2c). To further analyze these three subpopulations, we established threshold values to consider an astrocyte as GFP+ or CFP+ or both (i.e., STAT3+, NF-kB+ or

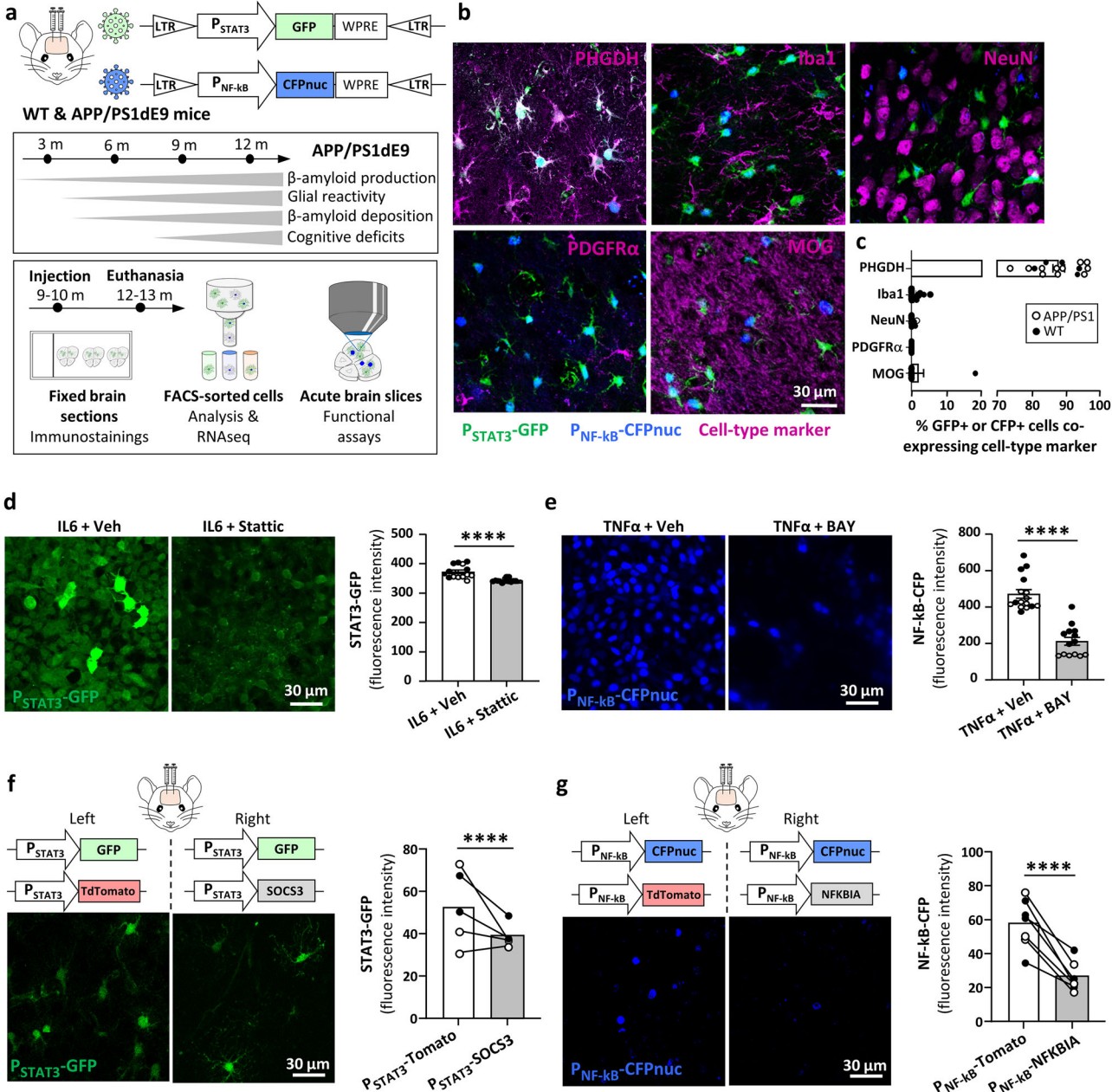

**Fig. 1 | Astrocyte-specific lentiviral reporters to monitor the STAT3 and NF-KB pathways in situ. a** Design of the two LV reporters for the STAT3 and NF-kB pathways and time line of typical pathological alterations in APP/PS1 mice[29,71]. GFP or nuclear CFP expression is under a minimal promoter with consensus responsive elements for STAT3 ($P_{STAT3}$) or NF-kB ($P_{NF-kB}$) respectively. Mokola-pseudotyped LV reporters are injected in the PFC of 10-month-old WT and APP/PS1 mice and analyzed 2 months later (unless specified otherwise) by fluorescence immunostainings, cytometry or functional assays on acute slices. **b** Representative images showing that the LV-$P_{STAT3}$-GFP and LV-$P_{NF-kB}$-CFPnuc reporters transduce PHGDH+ astrocytes, which express GFP (green) and/or nuclear CFP (blue), but not Iba1+ microglia, NeuN+ neurons, PDGFRα+ oligodendrocyte progenitor cells or MOG+ oligodendrocytes (cell-type markers in magenta). **c** Quantification of the total percentage of fluorescent cells (GFP+, CFP+ or both) co-expressing each cell-specific marker on the left. N = 11–21 mice. Exact N by marker and genotype in Source Data file. HEK cells were infected with LV-$P_{STAT3}$-GFP (**d**) or LV-$P_{NF-kB}$-CFPnuc (**e**), and treated after 48 h with 10 ng/ml IL6 (**d**) or TNFα (**e**) to stimulate the

corresponding pathway in presence of the pharmacological inhibitor Stattic (20 μM, **d**) or BAY 11-7082 (5 μM, **e**) or vehicle (Veh, DMSO). Both inhibitors significantly reduced reporter fluorescence. Linear mixed model (fixed effect: treatment; random effect: culture). ****$p < 0.0001$. N = 2 independent cultures, with n = 14–16 fields of view/condition. Exact n by condition in Source Data file. **f** The LV-$P_{STAT3}$-GFP reporter was injected in the mouse PFC with LV-$P_{STAT3}$-SOCS3 to inhibit the JAK-STAT3 pathway. GFP fluorescence intensity is decreased by 26% by SOCS3 compared to the control side injected with LV-$P_{STAT3}$-Td-Tomato. Td-Tomato/SOCS3, n = 1011/854 astrocytes; N = 5 mice. **g** LV-$P_{NF-kB}$-CFPnuc reporter was injected in the mouse PFC with LV-$P_{NF-kB}$-NFKBIA to inhibit the NF-kB pathway. NFKBIA decreases CFP fluorescence intensity by half compared to the control side. Td-Tomato/NFKBIA n = 1951/1675 astrocytes; N = 7 mice. **f, g** Linear mixed model (fixed effect: treatment; random effect: mouse; log-transformed data). ****$p < 0.0001$. **c, f, g** black and white circles represent WT and APP/PS1 mice, respectively. Lines represent measures taken in the two hemispheres of the same mouse. **d, e** black and white circles represent the two culture replicates.

STAT3+/NF-kB+ respectively, Fig. 2c). This analysis performed on several mouse cohorts, showed that STAT3+ and STAT3+/NF-kB+ subpopulations represented 31.7% and 45.7% of all fluorescent astrocytes respectively, while the NF-kB+ subpopulation was in minority,

representing 22.6% of the fluorescent cells (Fig. 2d). The three astrocyte subpopulations were also observed in similar proportion in a different β-amyloidosis model, the APP[NL-F/NL-F] mice[33], showing that this observation was not model-specific (Supplementary Fig. 1b). This cell

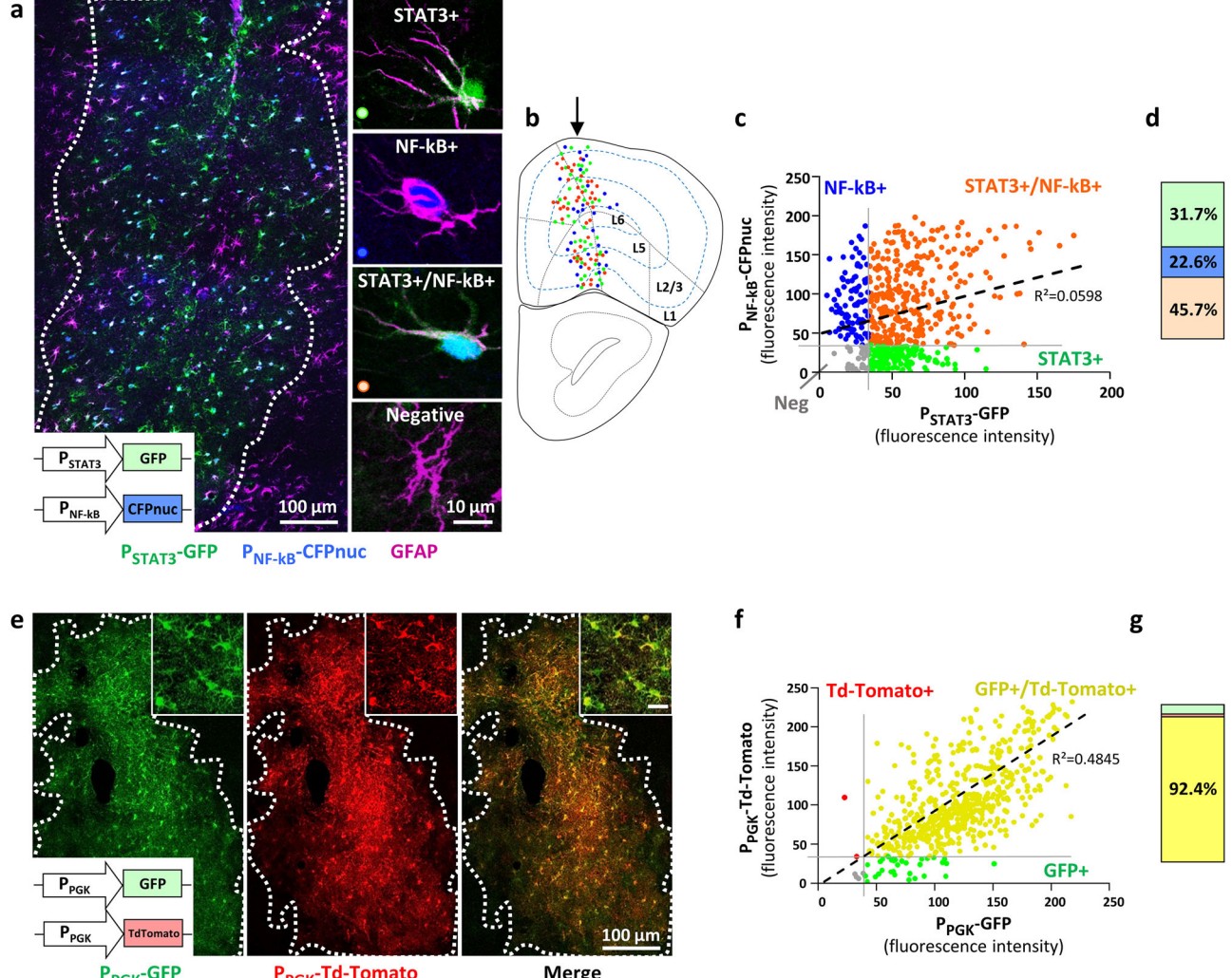

**Fig. 2 | Three astrocyte subpopulations evidenced by their active signaling pathways in the mouse PFC. a** Representative low (left) and high (right) magnification images showing astrocyte subpopulations in 12-month-old APP/PS1 mice: STAT3+ astrocytes expressing only GFP, NF-kB+ astrocytes expressing only CFP in the nucleus (CFPnuc), STAT3+/NF-kB+ astrocytes expressing both, plus a negative subpopulation of astrocytes labeled only by GFAP (magenta). The white dashed line demarcates the transduced site displaying fluorescent astrocytes. **b** Spatial mapping of the three fluorescent subpopulations on a PFC section of an APP/PS1 mouse (green dots: STAT3+ astrocytes, blue dots: NF-kB+ astrocytes and orange dots: STAT3+/NF-kB+ astrocytes). There is no specific topographical organization of any subpopulation across cortical layers. The arrow represents the injection site. Mouse brain section and cortical layers are taken form the Allen brain atlas. **c** Measurement of CFP and GFP fluorescent signal in single astrocyte soma on histological PFC sections. Three astrocyte subpopulations are defined according to GFP and CFP fluorescence thresholds (gray lines). Dot plot generated with all cells quantified from all mice. STAT3+ $n = 137$; NF-kB+ $n = 91$; STAT3+/NF-kB+ $n = 317$; $N = 7$. The

regression coefficient obtained is displayed on the graph. **d** Proportion of each astrocyte subpopulation, obtained from four independent cohorts with each $N = 6$–7 APP/PS1 mice, STAT3+ $n = 430$; NF-kB+ $n = 326$; STAT3+/NF-kB+ $n = 796$ quantified per subpopulation. **e** Representative low magnification images showing astrocytes co-expressing GFP and Td-Tomato after injection of two LV reporters with a ubiquitous promoter ($P_{PGK}$) in the PFC. The white dashed line demarcates the transduced site displaying fluorescent astrocytes. High magnification images are provided in insets, scale bar: 15 μm. **f** Measurement of Td-Tomato and GFP fluorescent intensity in single astrocyte soma. GFP and Td-Tomato levels are highly correlated and most cells are double-positive, according to the fluorescence thresholds shown as gray lines. Dot plot generated with 576 total representative astrocytes from $N = 6$ mice, to be comparable with panel c. The regression coefficient is displayed on the graph. **g** Proportion of astrocytes expressing GFP, Td-tomato or both. GFP+ $n = 134$; Td-Tomato+ $n = 21$; GFP+/Td-Tomato+ $n = 1872$; $N = 6$.

distribution was quite different from the one classically obtained following co-injection with two LV bearing the same ubiquitous promoter (here phosphoglycerate kinase promoter, $P_{PGK}$). In that case, transduced astrocytes displayed a correlated pattern of expression of the two reporter proteins (Fig. 2e, f), resulting in more than 92% of astrocytes co-expressing the two fluorescent proteins (Fig. 2g), which is expected given that LV are known to co-transduce cells, when injected in equal amounts[34,35]. This experiment confirms that the identified astrocyte subpopulations are not linked to stochastic transduction by LV, as astrocytes within the targeted PFC area are largely co-transduced by the two LV reporters. In addition, we

observed a high correlation of fluorescence levels between two different reporter proteins in astrocytes transduced with LV-$P_{STAT3}$-GFP and LV-$P_{STAT3}$-Td-Tomato or with LV-$P_{NF-kB}$-CFPnuc and LV-$P_{NF-kB}$-Td-Tomato in the PFC of APP/PS1 mice (Supplementary Fig. 1c–f). These results further show that this peculiar distribution of fluorescent astrocyte subpopulations is not caused by difference in half-life or fluorescent properties of the reporter proteins used.

To compare our results with data obtained in an unperturbed brain environment, we performed STAT3 immunostaining in uninjected Aldh1L1-eGFP mice that have GFP+ astrocytes, to assess astrocyte "intrinsic" ability to signal through the STAT3 pathway. We found

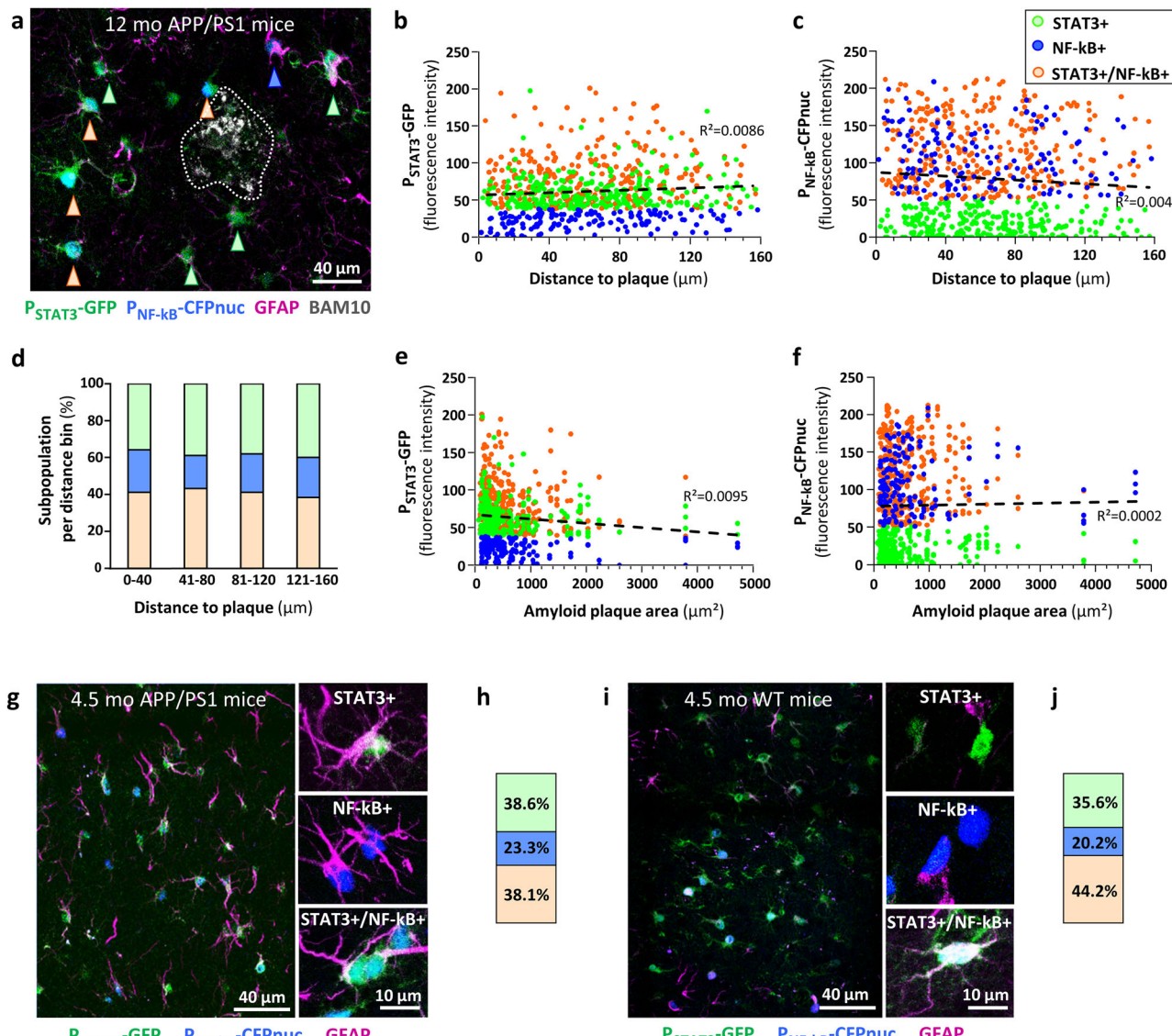

**Fig. 3 | STAT3 and NF-kB pathways activities are not impacted by the nearest amyloid plaque. a** Representative image from different mouse cohorts, showing astrocytes from the three subpopulations (highlighted with an arrow of the corresponding color) around BAM10+ amyloid plaques (white, delineated by dashes). Astrocytes are labeled with GFAP (magenta). STAT3 (**b**) and NF-kB (**c**) activities are not impacted by astrocyte distance to the nearest amyloid plaque. Regression analysis performed on all astrocytes, irrespective of their subpopulation, indicated by the usual color code. Note that the same analysis done without the negative astrocyte subpopulation in **b**, **c** respectively, gave a similar result. **d** The three astrocyte subpopulations are found in similar proportions, whatever the distance to amyloid plaques. STAT3 (**e**) and NF-kB (**f**) activities are not impacted by the area of the nearest amyloid plaque either. Regression analysis. **b**–**f**: STAT3+ $n = 275$; NF-kB+ $n = 150$; STAT3+/NF-kB+ $n = 303$; $N = 7$. Representative low (left) and high (right) magnification images (**g**) showing the three astrocyte subpopulations and their proportion (**h**) in 4.5-month-old APP/PS1 mice. STAT3+ $n = 339$; NF-kB+ $n = 204$; STAT3+/NF-kB+ $n = 334$; $N = 4$. Representative images (**i**) and astrocyte subpopulation proportion (**j**) in 4.5-month-old WT mice. STAT3+ $n = 301$; NF-kB+ $n = 171$; STAT3+/NF-kB+ $n = 374$; $N = 4$. The color code is displayed in the upper right part.

that not all PFC astrocytes express STAT3 at detectable levels, confirming that this transcription factor is present at variable levels among astrocytes, with a percentage of STAT3+ astrocytes (65%) aligning with the one measured with the LV reporter (Supplementary Fig. 1g). The same analysis was attempted with NF-kB, but none of the tested antibodies provided a reliable staining on PFC mouse sections.

In conclusion, our reporters allow the in situ detection of STAT3 and NF-kB activities and evidence three astrocyte subpopulations based on their activity for these cascades.

### Astrocyte subpopulations are not triggered by amyloid plaques

As amyloid plaques are known to impact gene expression in neighboring glia[36], we analyzed STAT3 and NF-kB activity in PFC astrocytes, depending on their proximity to amyloid plaques labeled with the BAM10 antibody. Astrocytes of the three subpopulations could be observed in close proximity to amyloid plaques, with no preferential accumulation of a specific subpopulation around plaques (Fig. 3a). We found indeed that neither GFP or CFP levels were impacted by astrocyte proximity to an amyloid plaque (Fig. 3b, c). Accordingly, the three subpopulations were observed with the same proportion whatever their distance to plaques, ruling out that they are less amenable to infection when in plaque proximity (Fig. 3d). The plaque size was also quantified and found to have no impact on GFP or CFP levels, and therefore on STAT3 and NF-kB pathway activity (Fig. 3e, f). Overall, this topographical analysis shows that the distribution of astrocyte subpopulations is not shaped by amyloid plaques.

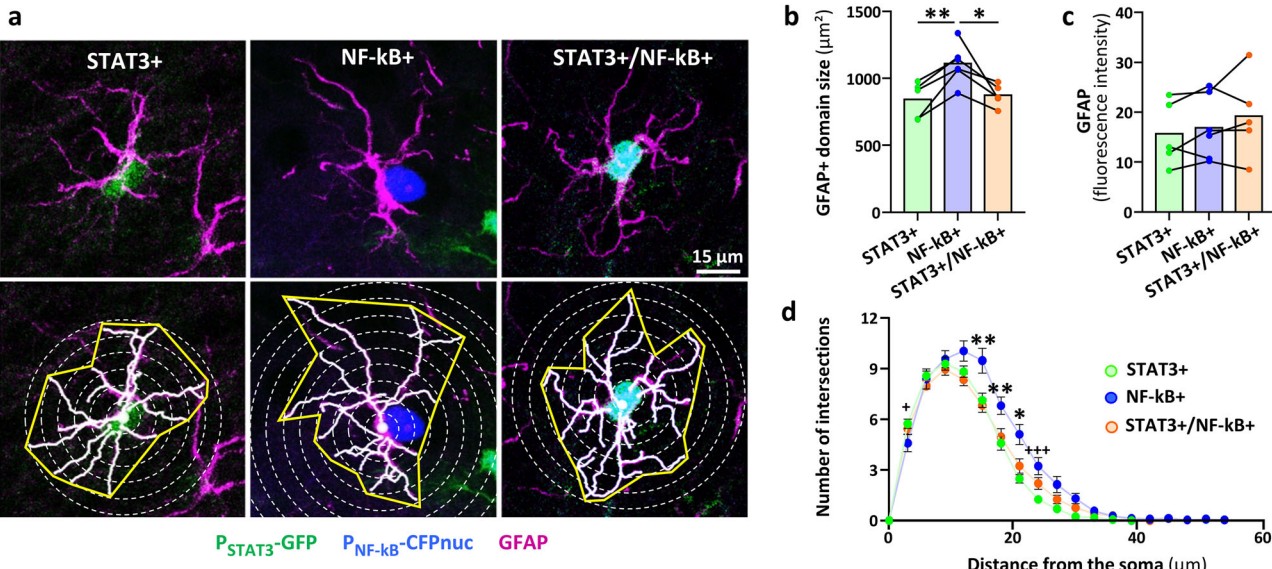

P$_{STAT3}$-GFP    P$_{NF-kB}$-CFPnuc    GFAP

**Fig. 4 | Astrocyte subpopulations display different domain size.**
**a** Representative images of an astrocyte of the three subpopulations in APP/PS1 mice (upper panel), and illustration of the 2D domain area measurement (yellow) and Sholl analysis (white circles, lower panel). Quantification of the GFAP+ domain area (**b**) and GFAP immunoreactivity within this domain (**c**) in the three subpopulations in 12-month-old APP/PS1 mice. NF-kB+ astrocytes are larger than STAT3+ and STAT3+/NF-kB+ astrocytes (**p = 0.003 and *p = 0.013, **b**) but express similar GFAP levels, p = 0.174 (**c**). Linear mixed model (fixed effect: subpopulation; random effect: mouse; log-transformed data for domain size analysis) and Tukey's

tests. The lines connect the three astrocyte subpopulations belonging to the same mouse. **d** Sholl analysis of the same astrocytes. Linear mixed model (fixed effects: subpopulation x distance; random effects: mouse and astrocyte nested in mouse; square-root-transformed data) and Tukey's tests. Stars indicate significance against both STAT3+ and STAT3+/NF-kB+ astrocytes, crosses indicate significance against STAT3+ astrocytes only. Exact p values are provided in Source Data file.
**b**–**d** STAT3+ n = 39 astrocytes, N = 5 mice; NF-kB+ n = 23, N = 6; STAT3+/NF-kB+ n = 36, N = 5.

To assess whether these subpopulations exist before plaque formation, we also studied younger APP/PS1 mice. Mice were injected with the LV reporters at 3.5-month-old and euthanized one month later, when amyloid plaques only start forming[37]. The three subpopulations were found in similar proportions in young and aged APP/PS1 mice (Fig. 3g, h), suggesting that these signaling astrocyte subpopulations exist before plaque formation. To further determine whether they are intrinsic to the mouse PFC, independently of an amyloid context, we injected the two reporters in young WT male littermates. We found the same three astrocyte subpopulations (Fig. 3i, j), in equivalent proportions.

These results show that PFC astrocytes display signaling heterogeneity in WT and APP/PS1 mice.

### Astrocyte subpopulations have different domain size
Our reporters allowed us to directly study these astrocyte subpopulations using a variety of quantitative and functional fluorescence-based methods (see Fig. 1a). We first analyzed the GFAP-based cytoskeletal structure of the three subpopulations in APP/PS1 mice, because both cascades have been shown to change the cytoskeleton and morphology of astrocytes in AD mice[16,19,22]. The 2D domain of NF-kB+ astrocytes was larger than those of STAT3+ and STAT3+/NF-kB+ astrocytes (Fig. 4a, b). Of note, GFAP immunoreactivity within the astrocyte domain was found equivalent between the three subpopulations, ruling out the fact that NF-kB+ astrocytes appear larger because they over-express GFAP (Fig. 4c). Sholl analysis also confirmed that NF-kB+ astrocytes were larger, as their Sholl curve was skewed to the right, while all three subpopulations displayed a similar maximal number of ramifications (Fig. 4d).

Expectedly, WT mice display lower GFAP levels than APP/PS1 mice in the PFC (see Fig. 3i), precluding a comparable analysis in these mice. Instead, we took advantage of electrophysiological recordings of the three astrocyte subpopulations in WT mice, to probe their intrinsic electrophysiological properties and use them as an indirect readout of

cell morphology[38]. The resting membrane potential was around −80 mV in the three subpopulations, as expected in astrocytes (Supplementary Fig. 2a). However, NF-kB+ astrocytes exhibited a significantly lower input resistance (Supplementary Fig. 2b) and increased membrane capacitance (Supplementary Fig. 2c), suggesting that NF-kB + astrocytes are larger than the other two astrocyte subpopulations in WT mice as well.

Overall, even if GFAP+ cytoskeleton labeling does not reveal the full complexity of astrocytes, our simple morphological analysis shows that the three astrocyte subpopulations defined by their active cascades differ by their domain size both in APP/PS1 and WT mice.

### Astrocyte subpopulations display molecular heterogeneity
Both STAT3 and NF-kB are transcription factors that regulate astrocyte molecular signatures. We thus performed RNA-seq analysis on the three astrocyte subpopulations isolated by FACS based on their GFP and CFP levels, from the PFC of 12-month-old APP/PS1 and WT mice (Supplementary Fig. 3a). DESeq2 analysis was first performed among the three astrocyte subpopulations of APP/PS1 mice. Despite low number of independent replicates per group, due to limited number of sorted astrocytes and stringent quality criteria applied (see Methods), we identified hundreds of differentially expressed genes (DEG) with high fold change between these three subpopulations in APP/PS1 mice (Supplementary Data 1). There were 635 DEG between STAT3+ and NF-kB+ astrocytes and 547 DEG between STAT3+ and STAT3+/NF-kB+ astrocytes, and none between NF-kB+ and STAT3+/NF-kB+ astrocytes (Fig. 5a, Supplementary Data 1). We focused on the list of 635 DEG between STAT3+ and NF-kB+ astrocytes in APP/PS1 mice, because they are two mutually exclusive subpopulations, and may thus display more tractable molecular differences. Among these 635 genes, 477 were expressed at higher levels in STAT3+ astrocytes than NF-kB+ astrocytes and 158 at lower levels (Fig. 5a, Supplementary Data 1). Transcription factor-based analysis with Pscan[39], which searches for consensus DNA motives on DEG promoter regions, identified NF-kB as a potential

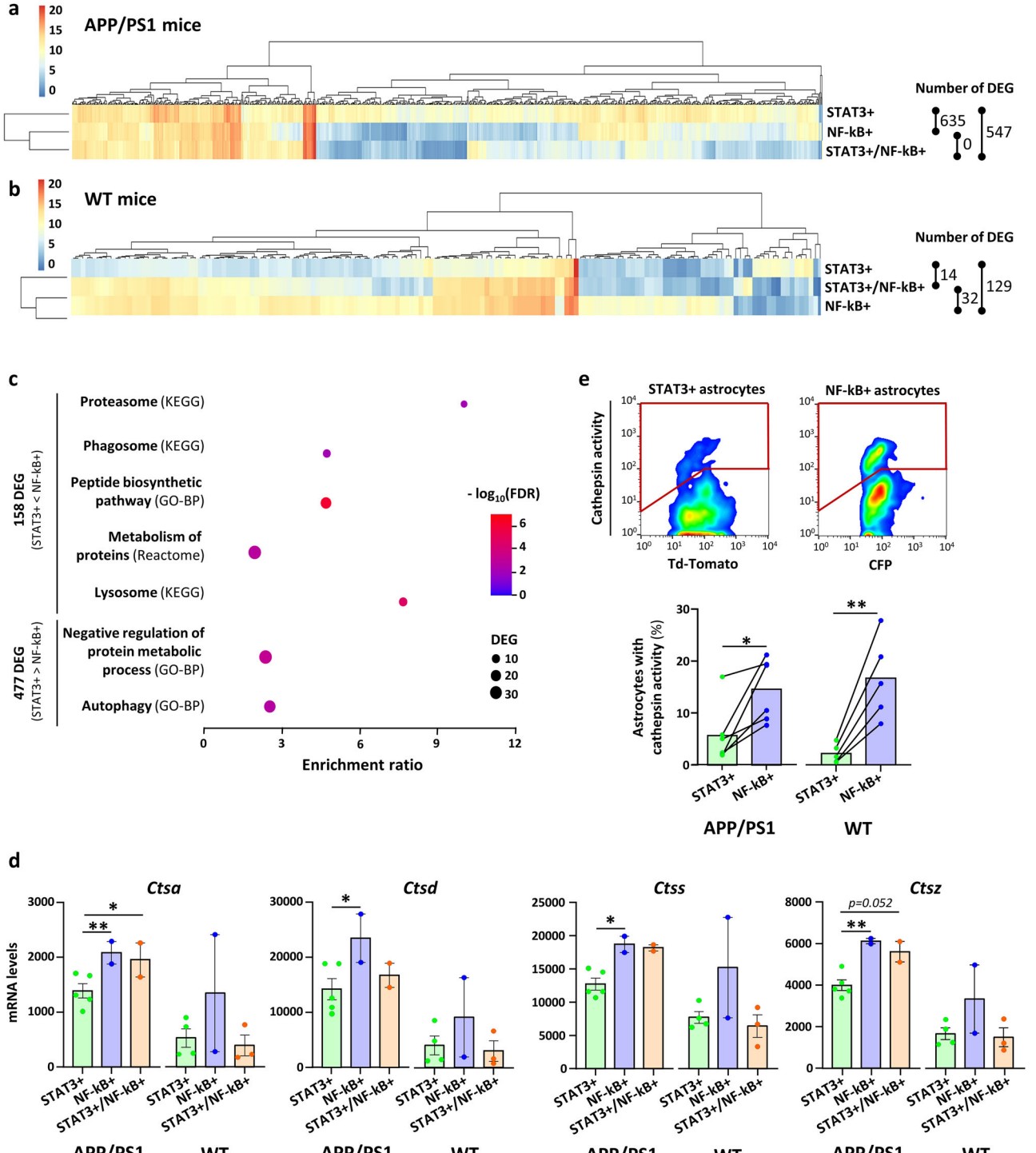

**Fig. 5 | FACS-sorted astrocyte subpopulations display specific molecular profiles and differential proteolytic capacity.** Heatmap and number of DEG between the three astrocyte subpopulations in APP/PS1 (**a**) and WT (**b**) mice. All unique DEG are shown (1065 and 145 unique DEG for APP/PS1 and WT mice, respectively). The scale represents the mean by subpopulation of expression counts transformed using the regularized log (rlog) method. **c** Dotplot showing Reactome, Gene Ontology (biological pathways, GO-BP) and KEGG terms linked to proteostasis (protein synthesis, recycling, degradation), identified by enrichment analysis on the 148 down- and 477 up-regulated genes between STAT3+ and NF-kB+ astrocytes in APP/PS1 mice. **d** Quantification of mRNA levels (transcripts per million, TPM) for four genes encoding lysosomal enzymes cathepsins (*Ctsa, Ctsd, Ctss,* and *Ctsz*) in

the three subpopulations of APP/PS1 and WT mice. *Ctsa* **p = 0.0067, *p = 0.0199; *Ctsd* *p = 0.0178; *Ctss* *p = 0.0108; *Ctsz* **p = 0.0032. N = 5 STAT3+ , 2 NF-kB+ and 2 STAT3+/NF-kB+ independent samples in APP/PS1 mice and N = 4 STAT3+ , 2 NF-kB+ and 3 STAT3+/NF-kB+ in WT mice. Statistical results from DEseq2 analysis. **e** FACS-analysis of cathepsin activity in astrocytes evidences a lower percentage of STAT3+ astrocytes with detectable cathepsin activity both in APP/PS1 and WT mice. Representative dotplots of a single APP/PS1 mouse (left: STAT3+ astrocytes, right: NF-kB+ astrocytes), quantifications are shown on the histogram below. The lines connect the two astrocyte subpopulations coming from the same mouse. N = 6 APP/PS1 and 5 WT mice. Two-way ANOVA (subpopulation, genotype) and paired *t*-test on arcsin-transformed data; *p = 0.017 and **p = 0.006.

master-regulator of the 158 genes significantly up-regulated in NF-kB+ astrocytes compared with STAT3+ astrocytes (Supplementary Table 1). An alternative analysis using chromatin immunoprecipitation databases (ChEA3-ENCODE) also identified NF-kB as the second most significant transcription factor (Supplementary Table 1). Conversely, STAT3 was identified as a master-regulator of the 477 DEG up-regulated in STAT3+ astrocytes by both methods (Supplementary Table 1).

Pathway analysis with KEGG, Gene Ontology-Biological Pathways and Reactome on the up- or down-regulated genes reported several enriched terms (Supplementary Data 2). Among them, several pathways linked to inflammation and response to stress were enriched in the list of DEG over-expressed in NF-kB astrocytes in APP/PS1 mice (Supplementary Data 2). NF-kB astrocytes also expressed higher levels of some inflammatory microglial genes (e.g., *Clec7a, CD68, Fcer1g*) as described in other pathological conditions[40,41], suggesting a more active and inflammatory profile in this subpopulation. We noted that the STAT3+ and NF-kB+ subpopulations also differed by several pathways linked to protein metabolism and clearance (i.e., proteostasis, Fig. 5c).

In WT mice, the comparison between STAT3+ and NF-kB+ subpopulations yielded a smaller list of 129 DEG (112 up and 17 down-regulated in STAT3+ astrocytes, Fig. 5a, Supplementary Data 1), resulting in only two significant KEGG pathways linked to metabolism (Supplementary Data 2). This observation suggests that molecular differences between STAT3+ and NF-kB+ astrocyte subpopulations are enhanced in a pathological context. To further explore this possibility, we compared the same astrocyte subpopulations between WT and APP/PS1 mice. There were 467 DEG between WT and APP/PS1 mice for STAT3+ astrocytes, 19 DEG for STAT3+/NF-kB+ and none for NF-kB+ astrocytes (Supplementary Data 1). Pathway analysis revealed that among the DEG down-regulated in STAT3+ astrocytes in APP/PS1 compared to WT mice, there were many terms linked to translation and oxidative phosphorylation, suggesting a global alteration in homeostatic processes in this subpopulation, caused by the pathological β-amyloidosis context (Supplementary Data 2).

Overall, our RNAseq analysis reports significant molecular differences between astrocyte subpopulations, driven by pathway activation and further enhanced by disease.

## Astrocyte subpopulations differ by their proteolytic activity

Given the pathological role of protein accumulation in AD, we decided to further explore the difference between STAT3 and NF-kB astrocytes regarding their intrinsic proteostatic function (Fig. 5c). Transcripts for four different lysosomal proteases cathepsins (*Ctsa, Ctsd, Ctss, Ctsz*, Fig. 5d) and catalytic subunits of the proteasome (*Psmb6*) or the immunoproteasome (*Psmb8, Psme2*, Supplementary Fig. 3b), were expressed at higher levels in NF-kB+ astrocytes than STAT3+ astrocytes in APP/PS1 mice.

To investigate whether these molecular differences translate into functional differences, we analyzed the proteolytic activity of the STAT3+ and NF-kB+ subpopulations using FACS-based functional assays. Cathepsin and proteasome activities were assessed with cell-permeant fluorescent activity probes in single living astrocytes, identified by their expression of reporter proteins on a cytometer (Supplementary Fig. 3c)[42]. Fewer STAT3+ astrocytes had detectable cathepsin activity than NF-kB+ astrocytes in APP/PS1 mice (Fig. 5e), in accordance with their lower expression of *Ctss* and *Ctsz*, which are specifically measured by this functional probe. The same significant difference between NF-kB+ and STAT3+ astrocytes was observed in WT mice (Fig. 5e). On the contrary, proteasome activity was found equivalent in STAT3+ and NF-kB+ astrocytes both in APP/PS1 and WT mice (Supplementary Fig. 3d).

Overall, this analysis shows that cortical astrocyte subpopulations exhibit tractable functional differences in their intrinsic proteolysis activity.

## Astrocyte subpopulations display differential hemichannel activity

Connexins are key proteins involved in two major astrocyte functions: gap junction coupling and hemichannel activity. We found that STAT3+ astrocytes expressed higher levels of the two main astrocyte connexins *Gjb6* (Cx30) and *Gja1* (Cx43) than the two other subpopulations in APP/PS1 mice (Fig. 6a). We confirmed that Cx30 is a downstream target of the STAT3 pathway at the protein level, by showing that SOCS3 reduced Cx30 immunoreactivity in the PFC of APP mice (Supplementary Fig. 4a). The pattern of Cx expression among astrocyte subpopulations was slightly different in WT mice: NF-kB+ astrocytes expressed significantly lower levels of *Gjb6* transcripts than both STAT3+ and STAT3+/NF-kB+ astrocytes, and a similar trend was observed for *Gja1* transcripts, without reaching significance.

We then assessed gap junction coupling by fluorescence recovery after photobleaching on acute slices incubated with sulforhodamine 101 (SR101), a fluorescent dye taken up by astrocytes that diffuses through gap junctions[43]. After SR101 loading, the soma of a single astrocyte was photobleached to monitor SR101 recovery through gap junctions with neighboring astrocytes (Supplementary Fig. 4b, c). We confirmed that SR101 fluorescent recovery was partially mediated by gap junctions, as it was significantly reduced by carbenoxolone, a broad-spectrum gap junction inhibitor (Supplementary Fig. 4b, d). In acute PFC slices of APP/PS1 mice injected with LV reporters, the speed of recovery and percentage of recovery at steady state did not differ between astrocyte subpopulations (Supplementary Fig. 4e, f), suggesting similar gap junction coupling.

Connexins also form hemichannels that mediate direct exchange with the extracellular space[44]. Hemichannel activity was assessed by measuring the uptake of the fluorescent dye ethidium bromide (EtBr) by astrocytes in acute brain sections[45]. We first confirmed that EtBr uptake was significantly reduced by carbenoxolone (Fig. 6b, c). Statistical analysis of EtBr uptake in the three astrocyte subpopulations in APP/PS1 and WT mice evidenced a significant genotype x subpopulation effect ($p < 0.0001$) (Fig. 6d, e), showing that differences in hemichannel activity among subpopulations are impacted by the mouse genotype. Specifically, in APP/PS1 mice, the STAT3+ subpopulation showed a significant ~40% higher hemichannel activity than both NF-kB+ and STAT3+/NF-kB+ subpopulations (Fig. 6d, e). In WT mice, NF-kB + astrocytes displayed significantly lower hemichannel activity than both STAT3+ and STAT3+/NF-kB+ astrocytes (Fig. 6e), in accordance with their lower Cx expression (Fig. 6a). Overall, the higher connexin expression in STAT3+ astrocytes, compared to NF-kB+ astrocytes in both WT and APP/PS1 mice, translates into higher hemichannel activity. In APP/PS1 mice, hemichannel activity is specifically reduced in STAT3+/NF-kB+ astrocytes.

Such differences in hemichannel activity between the three subpopulations in WT and APP/PS1 mice highlight both innate and β-amyloidosis-induced differences in astrocyte functional properties.

## Astrocyte subpopulation targeting impacts amyloid deposition and behavior

Finally, to interfere with astrocyte subpopulations in situ and assess their contribution to AD-related pathological symptoms, we inhibited these pathways specifically in astrocytes displaying an active cascade, by using our LV encoding STAT3 and NF-kB pathway inhibitors under their respective promoters (Fig. 1f, g). We compared four groups injected at 6-month-old: WT mice and APP/PS1 mice injected with the usual two LV reporters served as control groups (WT-control and APP-control groups), APP/PS1 mice injected with LV-P$_{STAT3}$-GFP and LV-P$_{STAT3}$-SOCS3 to block the STAT3+ subpopulation (APP-SOCS3 group) and APP/PS1 mice injected with LV-P$_{NF-kB}$-CFPnuc and LV-P$_{NF-kB}$-NFKBIA to target the NF-kB pathway (APP-NFKBIA group, Fig. 7a). Histological analysis of 11-month-old mice showed that astrocyte

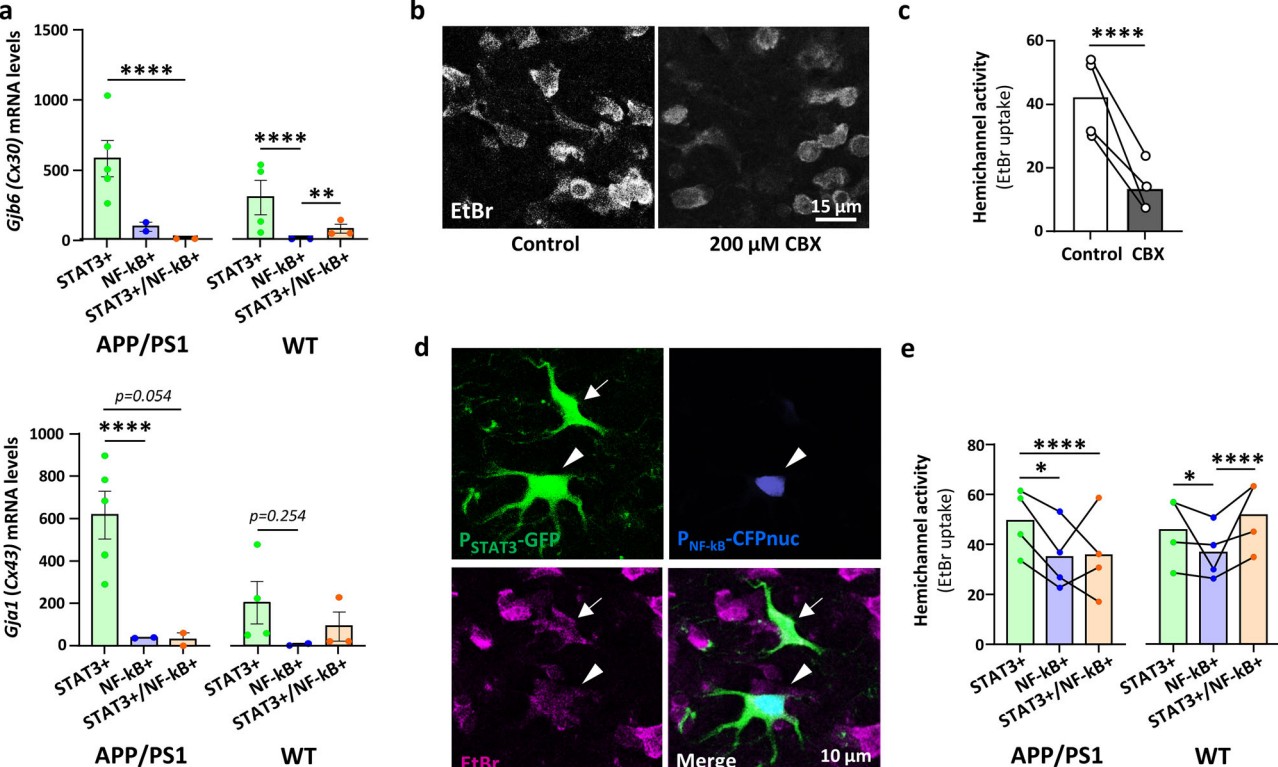

**Fig. 6 | STAT3+ astrocytes display high hemichannel activity with unchanged gap junction coupling. a** Quantification of mRNA levels of *Gjb6* (Cx30) and *Gja1* (Cx43) levels (TPM) in the three astrocyte subpopulations of APP/PS1 and WT mice. Statistical results are from DEseq2 analysis. ****$p < 0.0001$; **$p = 0.0032$. $N = 5$ STAT3+, 2 NF-kB+ and 2 STAT3+/NF-kB+ independent samples in APP/PS1 mice and $N = 4$ STAT3+, 2 NF-kB+ and 3 STAT3+/NF-kB+ in WT mice. **b** Representative confocal images of acute PFC slices showing EtBr (white) uptake in control conditions and after exposure to 200 μM carbenoxolone (CBX). **c** Quantification of EtBr uptake showing significant reduction by CBX. Control $n = 413$; CBX $n = 120$; $N = 4$. Linear mixed model (fixed effect: treatment; random effect: mouse),

****$p < 0.0001$. **d** Representative confocal images of acute PFC slices showing EtBr (magenta) uptake by a STAT3+/NF-kB+ (arrowhead) and a STAT3+ (arrow) astrocyte in an APP/PS1 mouse. **e** Quantification of EtBr uptake in the three astrocyte subpopulations in APP/PS1 and WT mice. STAT3+ $n = 99$; NF-kB+ $n = 70$; STAT3+/NF-kB+ $n = 158$; $N = 4$ in APP/PS1 mice; STAT3+ $n = 167$; NF-kB+ $n = 91$; STAT3+/NF-kB+ $n = 394$; $N = 4$ in WT mice. Linear mixed model (fixed effects: subpopulation x genotype; random effect: mouse nested in genotype), followed by Tukey's tests. *$p = 0.016$ and ****$p < 0.0001$ in APP/PS1 mice *$p = 0.022$ and ****$p < 0.0001$ in WT mice. Other subpopulation comparisons are not significant.

targeting did not significantly impact the density of BAM10-labeled amyloid plaques in the PFC of APP/PS1 mice (Supplementary Fig. 5a). However, APP-SOCS3 exhibited plaques with a significantly smaller median size than APP-control mice (−34%, Fig. 7b).

Mice were assessed on several tests involving the PFC[46], with alterations reported in APP/PS1 mice[47]. We evaluated anxiety on the elevated plus maze, short-term working memory on the spontaneous alternation test as well as social preference and social memory at the three-chamber test at 10-month-old. In the three tests, mice of the four groups efficiently explored the maze, covering a similar total distance, showing no alteration in locomotor activity (Supplementary Fig. 5b–d). At the elevated plus maze, APP-control and APP-SOCS3 mice spent significantly less time in the open arms than WT-control mice (Fig. 7c). APP-NFKBIA mice were not different from WT-controls, suggesting slightly improved anxiety levels (Fig. 7c). At the Y maze, all groups showed a similar percentage of spontaneous alternation, indicating no deficit in APP/PS1 mice and no effect of targeting different astrocyte subpopulations (Fig. 7d). Finally, at the three-chamber test, all groups showed the expected preference for the juvenile mouse over the object. However, mice in the APP-NFKBIA group spent significantly less time in contact with the juvenile mouse compared to the APP-control and the APP-SOCS3 groups (40% decrease, Fig. 7e). In the third phase of the test, when the test mouse was presented with the familiar and a novel juvenile mouse, only the WT-control and APP-control groups showed the expected preference for the novel juvenile, suggesting that mice in the APP-SOCS3 and APP-NFKBIA groups display altered

social memory (Fig. 7f). Overall, this behavioral analysis shows that targeting astrocyte subpopulations in the mouse PFC differentially impacts anxiety, social preference and memory.

The same experiment was performed in WT mice on an independent cohort. WT mice injected with LV-P$_{STAT3}$-SOCS3 or LV-P$_{NF-kB}$-NFKBIA in the PFC performed similarly at the elevated plus maze than control WT mice injected with both reporters (Supplementary Fig. 5e). At the three-chamber test, mice from the three groups showed the expected preference for the mouse over the object and for the novel over the familiar mouse (Supplementary Fig. 5f). Mice in the WT-NFKBIA group showed a reduction in total distance moved during the three phases of the three-chamber test (but not at the elevated plus maze), suggesting reduced mobility in this test, despite similar interest in the different stimuli presented (Supplementary Fig. 5g). Even if groups from the two independent cohorts cannot be directly compared, these results suggest that SOCS3 and NFKBIA influence mouse behavior differently in healthy and disease conditions.

In conclusion, we show that specific targeting of astrocyte subpopulations by inhibition of their upstream cascade impacts histological and behavioral AD-related alterations but has limited impact on WT mouse behavior, suggesting disease-specific roles.

## Discussion

We show, thanks to our astrocyte-specific LV reporters, that the STAT3 and NF-kB pathways signaling cascades define cortical astrocyte subpopulations characterized by signaling, morphological, molecular and

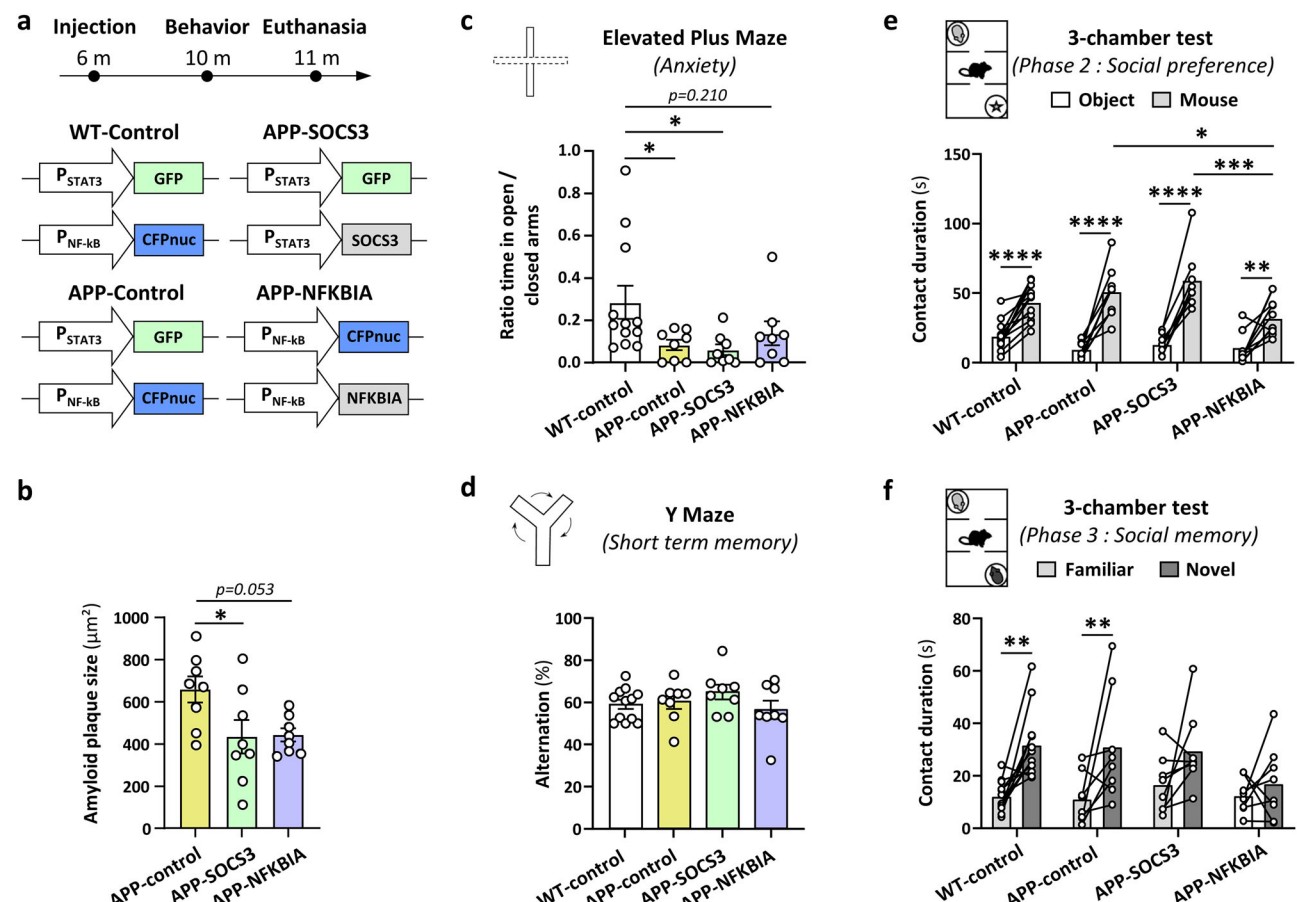

**Fig. 7 | Interfering with astrocyte subpopulations impacts amyloid deposition and behavior in APP/PS1 mice. a** Experimental design: two control groups (WT, $N = 12$; APP/PS1, $N = 8$) were injected at 6-month-old in the PFC with the classic combination of LV-$P_{STAT3}$-GFP and LV-$P_{NF-kB}$-CFPnuc reporters (WT-control group, APP-control group). Another group of APP/PS1 mice was injected with the LV-$P_{STAT3}$-GFP reporter and the LV-$P_{STAT3}$-SOCS3 (APP-SOCS3 group, $N = 8$) to interfere with the STAT3+ subpopulation. The last group of APP/PS1 mice was injected with the LV-$P_{NF-kB}$-CFPnuc reporter and LV-$P_{NF-kB}$-NFKBIA (APP-NFKBIA group, $N = 8$) to interfere with the NF-kB+ subpopulation. Behavioral testing was performed four months later, at 10-month-old and mice were euthanized at 11-month-old for histological analysis. **b** Quantification of the median size of BAM10-labeled amyloid plaques in the three APP/PS1 groups. APP-SOCS3 mice have significantly smaller amyloid plaques than APP-control mice (*$p = 0.043$). One-way ANOVA followed by Tukey's tests on square-root-transformed data. **c** Ratio of time spent in open over closed arms in the elevated plus maze. APP-control and APP-SOCS3 mice explore less the anxiogenic arm than WT-control (*$p = 0.049$ and 0.014 respectively). APP-NFKBIA are not different from WT-control ($p = 0.210$). The three APP/PS1 groups are not different from each other ($p > 0.676$). One-way ANOVA followed by Tukey's tests on square-root transformed data. **d** The percentage of spontaneous alternation in a Y maze is similar between groups. One-way ANOVA on arcsin-transformed data, $p = 0.361$. **e, f** At the three-chamber test, all groups prefer the juvenile mouse over the object (****$p < 0.0001$), but APP-NFKBIA mice spend significantly less time exploring the mouse compared to the APP-control and APP-SOCS3 mice (*$p = 0.028$ and ***$p < 0.001$ respectively) in the 2nd phase of the test (**e**). In the 3rd phase of the test, WT-control and APP-control groups spend significantly more time exploring the novel juvenile mouse (**$p = 0.001$ and 0.006 respectively), while APP-SOCS3 and APP-NFKBIA groups show no preference between the familiar and novel mouse, showing altered social memory (**f**). Two-way ANOVA (experimental group, stimulus) followed by Tukey's tests.

---

functional heterogeneity, which translates into differential impacts on AD-related outcomes in a mouse model.

Cell heterogeneity is typically assessed by single-cell or nucleus approaches or more recently, spatial transcriptomics. However, these are "terminal" methods that involve cell disruption or fixation and are thus incompatible with functional assays. Although single cell/nucleus transcriptomics allow the identification of several markers for specific cell clusters of interest, these marker panels require further validation and complex intersectional genetic techniques to label the targeted subpopulation and perform functional analysis. In contrast, our reporters enable direct, in situ identification of astrocyte states or subpopulations based on their cascade activity - a physiologically relevant readout. As they are fully compatible with multiple assays based on microscopy or cytometry, they allow direct probing of cell subpopulation intrinsic functions, instead of speculating based on gene ontology. It is well established that mRNA levels are often poor predictors of cell function[8,48] and indeed, we observed several

mismatches between mRNA expression profile and the actual functional features of astrocyte subpopulations in this study. For instance, we found that the lower proteasome subunit expression by STAT3+ astrocytes was not matched with an overall lower proteasomal activity in these cells. The fact that the proteasome is composed of more than thirty subunits, whose assembly and activity are tightly regulated at the post-transcriptional level[49] could explain this. Likewise, while we detected differential Cx levels between subpopulations, only hemichannel activity was found different, with no change in gap junction coupling, despite being operated by the same proteins. This further stresses the need for functional validation following molecular analysis[8,15]. The reporters also allow the direct identification of a major upstream transcription factor controlling the subpopulation, instead of inferring it from transcription factor analysis on DEG lists. By targeting the studied cascades with their cognate genetic inhibitor, we confirm that they are indeed involved in shaping astrocyte subpopulations in the PFC. Compared with exploratory omics approaches,

the main limitation of the reporter method is the analysis of only two pre-determined cascades, selected for their key role in astrocyte regulation. Indeed, our approach is blind to cells that do not signal through these two cascades, overlooking other sources of signaling heterogeneity. Our reporter-based approach could be applied to other important astrocyte signaling cascades, as well as other models or brain diseases.

We noted that the STAT3 reporter was less sensitive to pathway inhibition than the NF-kB reporter. This may be related to a lower efficacy of the pharmacological and genetic inhibitors used. Different mechanisms of action are at stake: NFKBIA directly binds NF-kB and sequesters it in the cytoplasm while SOCS3 acts upstream on the STAT3 pathway, by inhibiting receptor-bound-JAK[50]. The STAT3 reporter could still be activated by non-canonical STAT3 pathways, independent of JAK-mediated STAT3 phosphorylation[16,51,52].

The use of LV instead of transgenic reporter mice confers astrocyte specificity and enables the study of disease contexts by combining them with animal models. This may come at a price of some cell-to-cell variability in reporter protein expression. LV may also cause some local inflammation that was kept minimal by allowing at least one-month post-injection recovery and performing analysis at distance from the injection site. Despite these limitations, we were able to discriminate astrocyte subpopulations based on several independent structural, molecular and functional parameters. Whenever possible, we analyzed absolute GFP/CFP levels, instead of defining binary subpopulations (e.g., Figs. 1f, g, 2c, 3b, c, e, f), allowing a precise analysis of STAT3 and NF-kB activity levels. Binary classification may indeed fail to account for dynamic, subtle changes between astrocyte states, which could be better monitored with longitudinal two-photon microscopy. In the end, defining an astrocyte as STAT3+ does not imply an all-or-none ability, but rather an inclination or higher capacity for STAT3 signaling, which appears to be largely "hard-wired" and cell-intrinsic.

By studying the STAT3+/NF-kB+ subpopulation, we observed that depending on the specific gene or function considered, the STAT3 or NF-kB pathway dominated. For example in APP/PS1 mice, double STAT3+/NF-kB+ astrocytes have a similar domain size than STAT3+ astrocytes (Fig. 4b), but a gene expression profile for *Cts* closer to NF-KB+ astrocytes (Fig. 5d). Additional experiments will be required to fully explore the complex crosstalk between these two pathways and the molecular mechanisms involved.

It may seem surprising that neighboring astrocytes display variable levels of STAT3 and NF-kB pathways while being exposed to a similar extracellular milieu, and in the case of APP/PS1 mice, to a similar pathological environment. But in fact, there are few known examples of local signaling heterogeneity: only a subset of astrocytes exposed to spinal cord injury[53] or metastatic cells[54] are phospho-STAT3-positive. Neighboring cortical astrocytes can express or not the Gli1 transcription factor, suggesting differential ability to signal through the Sonic Hedgehog pathway[55], and only 20% of amygdala astrocytes display detectable levels of the oxytocin receptor[11]. Signaling heterogeneity was further evidenced functionally: different striatal astrocytes react to either D1 or D2 receptor-mediated neuronal activity[9] and a subpopulation of "learning associated astrocytes" encodes fear memory in the hippocampus, through the transcription factor Nuclear Factor I-A[56].

Such local heterogeneity is probably established during early brain development. Clonal analysis of cortical astrogenesis using the AstroBow[57] or Star-Track[58] fluorescent methods reveals intermingled astrocyte clones originating from distinct progenitors. This random, interwoven clone distribution is similar to the one we obtained for STAT3+, NF-kB+ and STAT3+/NF-kB+ subpopulations throughout PFC layers (Fig. 2b). Recent fate mapping experiments through genetic tagging also reveals five astrocyte subtypes dispersed throughout cortical layers at post-natal day 7[59]. Therefore, we could speculate that the signaling heterogeneity revealed by our reporters is established at

early developmental stages, probably through epigenetic regulation[60], resulting in astrocyte subpopulations with differential capacity to signal through one or the other cascade.

In an AD context, amyloid deposits could be expected to trigger further heterogeneity by creating hotspots of neuronal dysfunction and microglial activation, which in turn could locally activate STAT3 or NF-kB pathways. Indeed, astrocytes closer to amyloid plaques were shown to display stronger transcriptional changes[36,61] and morphological alterations[62,63] in AD mouse models and patients. Yet, we found no further induction of the two cascades in close proximity to amyloid plaques nor specific topographical organization of the subpopulations around them. This is another strong evidence for an intrinsic, hard-wired capacity of astrocyte subpopulations to signal via these cascades, independently of AD-related processes. In fact, recent single nucleus transcriptomic analysis in patients show that the identified clusters of astrocytes are already present in control subjects or at early disease stage[14], and rather display cluster-specific changes in gene expression[13]. Therefore, despite a heterogeneous environment around them, astrocytes keep their specific signaling capacity and most of their molecular profile. Even if different, the three astrocyte subpopulations retained typical astrocyte features such as astrocyte homeostatic gene expression, hyperpolarized resting membrane potential and gap junction coupling for example.

Our results support a model whereby pre-existing astrocyte subpopulations are further impacted by the β-amyloidosis context. Indeed, even if not directly patterned by amyloid plaques, astrocyte subpopulations were significantly influenced by the pathological environment with hundreds of genes repressed or induced among the STAT3+ subpopulation and NF-kB+ astrocytes displaying a more inflammatory molecular profile in APP/PS1 mice. Of note, NF-kB+ cells were found to express several microglial genes at higher levels than STAT3+ and STAT3+/NF-kB+ astrocytes in APP/PS1 mice. Detection of genes used as microglia markers in sorted astrocytes is not unexpected, as there are several reports of astrocytes expressing microglia-enriched genes in various pathological conditions[40,41] and even in physiological conditions[64,65]. The fact that other highly expressed microglial genes such as *Csf1r, Tmem119, Trem2* or *Cx3cr1* were not among the DEG suggests that NF-kB+ astrocytes acquire a specific inflammatory phenotype in APP/PS1 mice. Another example of pre-existing heterogeneity modulated in disease conditions is illustrated with hemichannels: astrocyte subpopulations display differential hemichannel activity in WT mice (i.e., NF-kB+ astrocytes have lower hemichannel activity) and hemichannel activity is reduced in STAT3+/NF-kB+ astrocytes of APP/PS1 mice, changing the overall pattern of functional differences among subpopulations in APP/PS1 mice. On the contrary, some subpopulation features are not affected in APP/PS1 mice, such as the larger size of the NF-KB+ subpopulation, observed in both genotypes despite GFAP induction in APP/PS1 mice. Overall, our results suggest that astrocytes may form distinct subpopulations based on their developmental origin and signaling capacity, but they also display significant plasticity in response to a changing environment.

We finally showed the differential impact of the three astrocyte subpopulations on AD-related alterations. We generated LV expressing an inhibitor under the pathway-sensitive promoter to create a negative regulatory feedback loop on the pathway in active astrocytes. We show that blocking the STAT3 pathway in astrocytes with SOCS3 induced social memory deficits in APP/PS1 mice. It also reduced the size of amyloid plaques deposited in the PFC. On the other hand, NFKBIA-inhibition of the NF-kB pathway in the PFC improved anxiety but it triggered deficits in social preference and memory. These effects were rather mild, which can be expected given that only a small fraction of the PFC was targeted with this LV strategy and the expression of the genetic inhibitor was not driven by a strong ubiquitous promoter. We can thus speculate that a stronger

or spatially extended pathway inhibition would have larger effects. Similar manipulation of astrocyte subpopulations in WT mice did not significantly affect these behaviors, showing that the identified subpopulations have distinct effects on behavior only in a disease context. The current study identifies a positive impact of STAT3+ astrocytes in the PFC, as their inhibition triggers memory deficits in APP/PS1 mice. This is in contrast with the improved spatial memory observed after SOCS3 overexpression in the dorsal hippocampus[19], which further illustrates region-specific roles for astrocytes in controlling behaviors in health and disease[66,67].

Of note, our strategy based on STAT3 inhibition in astrocytes with SOCS3 will inevitably affect the STAT3+/NF-kB+ subpopulation as well. Instead of depleting the STAT3+ subpopulation selectively, we rather abolish the STAT3+ subpopulation and transform the STAT3+/NF-KB+ subpopulation into a single NF-KB+ subpopulation (and the same reasoning applies to NFKBIA effects). We thus blunted astrocyte signaling heterogeneity, at least regarding these two cascades. In spite of this limitation, our results show that astrocyte subpopulations are important for brain function in pathophysiological conditions, as their targeting impaired specific mouse behaviors in APP/PS1 mice.

Inhibition of the STAT3+ subpopulation, characterized by a unique proteostatic profile with low cathepsin activity, resulted in smaller amyloid plaques in the PFC, suggesting that the STAT3+ subpopulation may promote amyloid plaque expansion. The mismatch between SOCS3 positive effects on amyloid deposition and negative effects on social memory is not unexpected. Amyloid plaques are just a terminal index of AD pathogenesis, while mouse behaviors are dynamically regulated by complex interactions with neurons and may be altered even before plaque deposition[68]. For example, STAT3+ astrocytes could regulate neuronal activity in the PFC through gliotransmitter release, possibly through their higher hemichannel activity[69,70]. The functional impact of the three astrocyte subpopulations on neuronal circuits is however, quite challenging to test experimentally, given the lack of methods to probe gliotransmitter release by single cells.

Overall, our study takes advantage of versatile reporters to study cortical astrocyte signaling heterogeneity. We find that astrocyte subpopulations co-exist in the cortex of WT and APP/PS1 mice, defined by their signaling ability. They exhibit unique morphological, molecular, and intrinsic functional profiles, with differential impacts on AD-related symptoms, identifying their upstream regulatory cascades as potential modulators of disease progression.

## Methods
### Animals
Experiments were carried out on male APP/PS1dE9 (APP/PS1) mice[29] and their wild-type (WT) littermates on the C57BL/6 J background. These mice express a chimeric mouse/human APP695 gene containing the Swedish K670M/N671L mutations and human *PSEN1* with a deletion in exon 9, and display progressive histological and behavioral symptoms (Fig. 1a)[29,71]. We also used 7-month-old Aldh1L1-eGFP mice on the C57BL/6J background to identify GFP+ astrocytes and 6-month-old APP[NL-F/NL-F] mice[33] on a C57BL/6 J background as an additional β-amyloidosis model. All mice were housed under standard environmental conditions (12-hour light-dark cycle, temperature 22 +/−1 °C, and humidity 50%) with *ad libitum* access to food and water. Mice were analyzed around 12 months, except when mentioned otherwise. All procedures were reviewed and approved by a local ethics committee (CETEA N°44), and by the French Ministry of Education and Research (APAFIS # #33827-2021110914102549 v4 and #4565-20 16031711426915 v3). They were performed in an authorized animal facility (#D92-032-02), in strict accordance with recommendations of the European Union (2010-63/EEC), and in compliance with the 3 R guidelines. Animal care was supervised by a dedicated veterinarian and animal technicians.

### Lentiviral vectors
Astrocyte-specific lentiviral vectors (LV) encoding reporters were generated to monitor the activity of the transcription factors STAT3 and NF-kB (p65). Six copies of the *cis*-acting enhancer element recognized by STAT3 or NF-kB were placed upstream of a minimal TA promoter composed of the TATA box from the Herpes simplex virus thymidine kinase promoter (as described in plasmids #LR0077 and LR0051 from Panomics, respectively). The STAT3-responsive promoter ($P_{STAT3}$) drives the expression of the fluorescent proteins eGFP or Td-Tomato, while the NF-kB-responsive promoter ($P_{NF-kB}$) drives the expression of eCerulean with a nuclear localization sequence (CFPnuc) or Td-Tomato (Fig. 1a). The woodchuck post-regulatory element (WPRE) was inserted downstream of the transgenes to enhance their expression. LV reporters were pseudotyped with the Mokola envelope to promote astrocyte tropism[28] (Fig. 1b). These reporters are later called LV-$P_{STAT3}$-GFP, LV-$P_{STAT3}$-Td-Tomato, LV-$P_{NF-kB}$-CFPnuc and LV-$P_{NF-kB}$-Td-Tomato.

We also generated astrocyte-specific LVs encoding SOCS3 or NFKBIA under $P_{STAT3}$ or $P_{NF-kB}$ respectively (LV-$P_{STAT3}$-SOCS3 and LV-$P_{NF-kB}$-NFKBIA, respectively) to inhibit the two pathways in astrocytes. Additional LVs used in this study include astrocyte-specific LVs encoding Td-Tomato or GFP under a constitutive phosphoglycerate kinase (PGK) promoter (named LV-$P_{PGK}$-Td-Tomato, LV-$P_{PGK}$-GFP) to measure the rate of co-transduction in PFC astrocytes.

### Lentiviral vector microinjections
Mice were anesthetized with an *i.p.* injection of ketamine (100 mg/kg) and medetomidine (0.25 mg/kg) and anesthesia was reversed by a *s.c.* injection of atipamezole (0.25 mg/kg) at the end of the surgical procedure. Mice were given a *s.c.* injection of buprenorphine (0.075 mg/kg) and xylocaine (5 mg/kg, at the incision site) 5 min prior to surgery. Mouse temperature was maintained at 37 °C throughout the procedure, thanks to a heating pad connected to a rectal temperature probe. Mice were given *s.c.* injection of 100 μl of warm saline before and after surgery to prevent dehydration.

Viral vectors were microinjected in the mouse PFC (ventral and lateral orbital cortex). Coordinates from Bregma were: anteroposterior: +2.8 mm, lateral: +/−1 mm; ventral: −2.25 mm from the skull surface, measured at Bregma, with the tooth bar set at 0.0 mm. The two LV reporters (or a LV reporter and LV-$P_{STAT3}$-SOCS3 or LV-$P_{NF-kB}$-NFKBIA for Figs. 1f, g, 7) were diluted in PBS with 1% bovine serum albumin (BSA), at a final concentration of 50 ng p24/μl each and 2 μl of the dilution was injected at a rate of 0.2 μl/min with a pump. Mice were injected 1–3 months before analysis, except for subpopulation targeting with SOCS3 and NFKBIA (Fig. 7).

### Fluorescent immunostainings
Mice were euthanized with an *i.p.* injection of buprenorphine (0.3 mg/kg) and a mix of ketamine (330 mg/kg) and medetomidine (1 mg/kg). They were then transcardially perfused with PBS for 2 min, and with 4% paraformaldehyde (PFA) for 8 min at 8 ml/min. Brains were then removed and post-fixed in 4% PFA overnight at 4 °C. Brains were cryoprotected in 30% sucrose solution for 48 h at 4 °C and cut into 30 μm serial sections with a freezing microtome. Sections were stored in 30% ethylene glycol, 30% glycerol, 40% PB at −20 °C until use.

For immunostainings, slices were rinsed 3 × 10 min in PBS and blocked in 4.5% normal goat or horse serum (NGS or NHS, Invitrogen) or in 3% BSA (Sigma) depending on the antibody used, diluted in PBS with 0.2% Triton X-100 (PBST) for 1–2 h at room temperature. Slices were then incubated with primary antibodies diluted in PBST with 3% NGS, NHS or BSA for 24 or 48 h at 4 °C. The following antibodies were used: mouse anti-BAM10 (Sigma, A3981, 1:500), rabbit anti-Cx30 (Invitrogen, 71-2200, 1:500), mouse anti-GFAP-Cy3 (Sigma, C9205, 1:1,000), chicken anti-GFP (Aves labs, GFP-1020, 1:1,000) rabbit anti-Iba1 (Wako, 019-19741, 1:1,000), mouse anti-MOG (Millipore,

MAB5680, 1:500), chicken anti-NeuN (Sigma, ABN91, 1:1,000), rabbit anti-PDGFRα (Cell Signaling, #3174, 1:500), guinea pig anti-PHGDH (Frontier, 3PGDH-GP-Af198, 1:250), rabbit anti-STAT3a (Cell Signaling, 8768 P, 1:500) and goat anti-Td-Tomato (Sicgen, AB818-200, 1:1,000). After 3 × 10 min rinses, slices were incubated for 1 h with the appropriate AlexaFluor conjugated secondary antibody produced in goat or in donkey and diluted in PBST with 3 % NGS, NHS or BSA. Slices were rinsed, mounted on microscope glass slides with Vectashield mounting medium (Vector laboratories) and coverslipped. For STAT3 immunostaining, an antigen retrieval step was included before the blocking step: slices were incubated for 20 min at −20 °C in pre-cooled 100% methanol before 3 × 10 min washes in PBS. STAT3a antibody was diluted in SignalStain Antibody Diluent (Cell Signaling, 8112) and incubated 72 h at 4 °C.

### Image acquisition and analysis

Images were acquired with an inverted laser scanning confocal microscope (TCS SP8, Leica), controlled by the LAS X software (Leica). Tiled and z-stacked confocal images (10 z-steps of 1 μm, kept constant within cohort, maximum intensity stack) of the full PFC displaying fluorescent cells were acquired on 3–6 sections per animal with a 40X objective, at a speed of 400 Hz (pixel size = 0.569 μm/resolution: 512 × 512). Exposure time and laser power were optimized within each analysis to avoid saturation based on fluorescence intensity histograms. CFP and GFP were always acquired on separate sequences using the following parameters that avoid crosstalk between fluorescent proteins: Emission range in nm for CFP: 455–480; GFP: 493–540 nm, with excitation at 405 and 488 nm respectively. Two to three scans per field of view were acquired and averaged to improve signal over noise. These parameters were kept constant for all mice and sections within an analysis, performed with the FIJI software. To validate LV tropism, the percentage of CFP+ and/or GFP+ cells co-expressing different cell-type specific markers was quantified from a minimum of 3 field images per section and 3–6 sections per mouse. For other analysis of astrocyte subpopulations, visible GFP+, and/or CFP+ soma (or Td-Tomato+ for some experiments) were manually delineated from a minimum of 3 images per section and 3–6 sections per mouse. The mean gray value of each cell was extracted with the FIJI software, background fluorescence intensity was subtracted for each channel and the specific fluorescence intensity was plotted as dotplots. Only when necessary, a positive threshold was established for each fluorescent protein to define three subpopulations and study their proportion and properties (e.g., GFAP+ domain, distance to plaque). To analyze STAT3 expression in PFC astrocytes of Aldh1L1-eGFP mice, the percentage of STAT3+/GFP+ astrocytes and STAT3-/GFP+ was manually calculated from 6 to 8 fields of view per mouse. For the topographical analysis relative to BAM10+ amyloid plaques, the closest amyloid plaque was identified and manually delineated on FIJI to measure its size. The distance between this amyloid plaque and each type of astrocyte (determined by its level of GFP and CFP fluorescence, as described) was then measured from the center of the astrocyte soma to the edge of the closest amyloid plaque. The measurement of the 2D astrocyte total GFAP+ domain area and Sholl analysis were performed on individual astrocytes on image stacks, avoiding astrocytes in direct contact with plaques as they are usually very polarized and with abnormal morphology. The GFAP+ domain was measured by connecting the terminal tips of each GFAP+ process. For Sholl analysis, the number of intersections between GFAP+ processes and concentric circles of 3 μm around the soma was measured using the FIJI SNT framework of the neuroanatomy plugin. Cx30 immunoreactivity was measured in 4–6 fields of view per mouse, acquired with the confocal microscope at a 40X magnification, within the targeted, fluorescent PFC region.

If a mouse brain displayed signs of local neuroinflammation due to the surgery (major increase in GFAP or Iba1 expression, excessive glial scar at the site of injection), it was excluded from the study (max.

1–2 mice per cohort). In all cases, the local glial scar visible at the site of injection was excluded from the analysis.

### Fluorescent activated cell sorting for RNA sequencing analysis

Twelve-month-old WT and APP/PS1 mice were killed by cervical dislocation and their PFC rapidly collected in Hank's Balanced Salt Solution (HBSS) without $Ca^{2+}$ and $Mg^{2+}$ (Sigma). The PFC of three mice were pooled, resulting in $N = 4$ and 5 independent PFC samples in WT and APP/PS1 mice respectively. Cells were mechanically and enzymatically dissociated with fire-polished Pasteur pipettes and the neural tissue dissociation kit with papain (Miltenyi Biotec), following manufacturer's instructions. Myelin removal beads II (Miltenyi Biotec) were used to deplete myelin from cell suspensions using MS columns. Cells were centrifuged at $300 \times g$ for 5 min at 4 °C, resuspended in 400 μl HBSS with $Ca^{2+}$ and $Mg^{2+}$ and sorted on a BD Influx cell sorter (BD biosciences). GFP was detected at 530/40 nm (488 nm excitation) and CFP at 450/50 nm (408 nm excitation). Control samples of unlabeled or mono-fluorescent brain cells were used to define detector gains and sorting gates, which were kept constant for all samples. No compensation was required. Cells were gated on a side scatter/ forward scatter plot, then singlets were selected and GFP+, CFP+ and GFP+/CFP+ astrocytes were sorted (Supplementary Fig. 3a). Cells were centrifuged at $300 \times g$ for 5 min at room temperature, lyzed in 400 μl TRIzol (Invitrogen) and stored at −80 °C before RNA extraction with on-column DNAse digestion, according to manufacturer's instructions (RNeasy micro kit, Qiagen). RNA was eluted in 14 μl of RNAse-free deionized water and stored at −80 °C before transcriptomic analysis.

### RNA sequencing and analysis

The protocol described in ref. 19 was used to analyze the transcriptome of the three astrocyte subpopulations, except that the lower number of astrocytes sorted per sample was not compatible with RNA quality assessment, and full-length double strand cDNA libraries were amplified with 18 LD-PCR cycles. RNAseq libraries were sequenced on a Novaseq Illumina 6000 platform (2 × 100 bp). Quality control of sequencing data was performed with FastQC (v0.11.9). Reads were mapped on the GRCm39 (mm39) mouse genome assembly with STAR (v2.7.9a) and quantified with RSEM (v1.3.3). We selected samples that passed quality control filters based on the number of detected genes (>15% expressed coding genes; threshold at one transcript per million, TPM) and the number of uniquely mapped reads (>60%). This excluded nine samples, probably due to the small number of input cells. The final number of independent samples per group was $N = 5, 2, 2$ for STAT3+, NF-kB+ and STAT3+/NF-kB+ astrocytes in APP/PS1 and $N = 4, 2, 3$ in WT mice. Differential gene expression analysis was performed with DESeq2 (v1.32.0) Bioconductor (v3.13) package on R (v4.1.1). Only genes with a raw count number ≥10, in at least one sample in each of the two compared groups were analyzed. Results were considered statistically significant for an adjusted $p \le 0.05$ and a $|\log2(\text{fold-change})| \ge 1$. Transcript levels across astrocyte subpopulations are displayed as TPM. Non-fluorescent cells were not analyzed because they were composed of very heterogeneous cells (microglial cells, neurons, oligodendrocyte progenitor cells but also many non-transduced, non-fluorescent astrocytes), and thus it was not possible to compute an enrichment score for astrocyte markers over other cell type markers. To identify putative upstream transcription factors of DEG, we used Pscan (v. 1.6)[39] and ChEA3 online tool (https://maayanlab.cloud/chea3/ - accessed in June 2025). Pscan is based on motif recognition in the −450 to +50 bp promoter region of genes of interest, using the TRANSFAC database of 282 potential transcription factors (v. 7.0 – public release). We report the statistics obtained with DNA motives matching the sequence present in our LV reporters (i.e STAT3_02 and NFKAPPAB_01). The ChEA3-online tool explores human ChIP-seq data from the ENCODE project. Reactome, Gene Ontology-Biological Process and KEGG pathway enrichment analysis were

carried out with WebGestalt v.2024 (https://www.webgestalt.org/ - accessed in June 2025) using the false discovery rate (FDR) statistics.

## Fluorescence activated cell sorting analysis of proteolytic activity

For this assay, LV-P$_{STAT3}$-Td-Tomato was used instead of LV-P$_{STAT3}$-GFP to be compatible with the proteasome fluorescent probe. These two STAT3 reporters yielded similar results in situ (Supplementary Fig. 1c, d). APP/PS1 and WT mice were injected with the LV-P$_{STAT3}$-Td-Tomato and LV-P$_{NF-kB}$-CFPnuc reporters. They were euthanized by cervical dislocation and their PFC rapidly collected in HBSS and dissociated as described in section "*Fluorescent activated cell sorting for RNAseq analysis*". Cells were then resuspended in 0.5% PNB buffer (Perkin Elmer, FP1020) and incubated for 30 min with 1 μM fluorescent cathepsin probe (iABP, Vergent Bioscience, 40200, which more specifically detects the activity of CSTB, L, S and X/Z) and 200 nM proteasome probe (UbiQ, Bio BV, UbiQ-018) at room temperature. These cell permeable activity probes become fluorescent when processed by cathepsin or the proteasome, respectively[42]. Cells were centrifuged at $300 \times g$ for 5 min at 4 °C and resuspended in 400 μl HBSS before analysis on a BD Influx cell sorter. CFP was detected at 460/50 nm (408 nm excitation), the proteasome probe at 530/40 nm (488 nm excitation), Td-Tomato at 579/34 nm (561 nm excitation) and the cathepsin probe at 670/30 nm (646 nm excitation). Control samples of unlabeled, mono-fluorescent and fluorescent-minus-one samples brain cells were used to define detector gains and sorting gates, which were kept constant for all samples. No compensation was needed to quantify the activity of proteasome and cathepsins as well as CFP and Td-Tomato levels, which were excited by four different lasers. FACS data was analyzed with FlowJow (V10). Cells were gated on a side scatter/forward scatter plot, then singlets were selected and CFP+ or Td-Tomato+ astrocytes were selected and finally the percentage of cells within each subpopulation, which had a detectable cathepsin or proteasome activity was calculated, thanks to gates defined in control samples incubated without the probe (Supplementary Fig. 3c). Of note, double CFP+/Td-Tomato+ astrocytes overlapped with autofluorescent cells in the tail of the major negative cell population and could not be analyzed reliably.

## Acute PFC slice preparation for gap junction coupling and hemichannel analysis

Mice were euthanized by cervical dislocation. After rapid brain removal from LV-injected mice, acute 300 μm coronal PFC slices were sectioned with a vibratome (Leica VT1200S) in ice-cold oxygenated solution (95% O$_2$/5% CO$_2$) containing (in mM): NaCl (87), NaHCO$_3$ (25), sucrose (75), dextrose (10), KCl (2.5), NaH$_2$PO$_4$-H$_2$O (1), MgCl$_2$-6H$_2$O (7), CaCl$_2$ (0.5). For recovery, slices were transferred for 1 h at room temperature into oxygenated artificial cerebrospinal fluid (aCSF) containing (in mM): NaCl (119), NaHCO$_3$ (26.2), dextrose (11), KCl (2.5), NaH$_2$PO$_4$-H$_2$O (1), MgSO$_4$-7H$_2$O (2), CaCl$_2$ (2.5).

## Fluorescence recovery after photobleaching (FRAP)

Gap junction coupling between astrocytes was assessed with sulforhodamine 101 (SR101), an astrocyte-specific fluorescent dye that diffuses through gap junctions. Acute PFC slices from APP/PS1 mice were incubated in 1 μM SR101 in aCSF for 20 min at 37 °C. After a 20 min wash in aCSF, slices were placed in a perfusion chamber under an Axio Examiner Z1 two-photon microscope (Zeiss) with a 40X water-immersion objective (NA 1.0) and perfused with oxygenated aCSF at 5 ml/min with a peristaltic pump throughout the experiment. SR101+, GFP+ and/or CFP+ astrocytes were identified using two-photon illumination (3i Intelligent Imaging) from a Chameleon Ultra II Ti-sapphire laser (Coherent). SR101 was excited at 900 nm and fluorescence was detected by a photo-multiplier tube with a 616/69 nm emission filter and a 580 nm dichroic mirror. GFP was excited at 900 nm and fluorescence detected by a photo-multiplier tube with a 525/40 nm

emission filter and a 580 nm dichroic mirror. CFP was excited at 800 nm and fluorescence was detected by a photo-multiplier tube with the 525/40 nm emission filter and a 580 nm dichroic mirror. FRAP experiments consisted in the following illumination sequence[43]: (i) after identifying an astrocyte positive for SR101 and either one or both fluorescent reporters, its SR101 baseline fluorescence was measured at 1 Hz for 30 s; (ii) a two-photon light pulse (2 s, 450–550 mW) was applied at 550 nm to photobleach SR101 in the astrocyte soma; (iii) SR101 fluorescence recovery was recorded at 1 Hz for 3 min. Images were analyzed with ImageJ software. Baseline intensity was averaged over the first 30 s of acquisition and set to 100% for each astrocyte. Fluorescence recovery was calculated as the percentage difference between the average fluorescence during the last 5 imaging time-points, and the lowest fluorescence (measured following photo-bleaching), divided by the percentage of photobleaching (Supplementary Fig. 4c). The speed of recovery S was obtained by fitting the recovery part of the curve with the following equation: *Normalized fluorescence = S x ln(time) + b* (Supplementary Fig. 4c). All experiments were carried out at a depth of 70–100 μm below the slice surface to avoid damaged cells caused by sectioning. Two to ten astrocytes per slice were analyzed, in 2–3 slices per mouse.

## Ethidium bromide uptake

To assess hemichannel activity, we used ethidium bromide (EtBr), a fluorescent agent that enters cells through active hemichannels[45]. Acute PFC slices from WT and APP/PS1 mice were incubated 15 min at 37 °C in oxygenated aCSF or with 200 μM carbenoxolone in oxygenated aCSF. Slices were incubated for 20 min at 37 °C with 4 μM EtBr in oxygenated aCSF and washed for 20 min in aCSF. Sections were then fixed in 4% PFA in PBS for 1 h at 4 °C, rinsed in PBS and analyzed on a confocal microscope with a 40X objective. EtBr fluorescence intensity was measured in a single focal plane (1 μm) of single astrocytes with the ImageJ software. Quantifications were performed within 15 μm around the center of each slice to avoid surface artefacts. The background fluorescence signal was measured in each image and subtracted to EtBr intensities measured in astrocytes of the same image. GFP and CFP intensity were also measured in each astrocyte to assign them to one of the three subpopulations.

## Ex vivo electrophysiological recordings

LV-injected mice were euthanized by cervical dislocation. After rapid brain removal, acute 325 μm coronal PFC slices were sectioned with a vibratome (Leica VT1200S) in ice-cold aCSF saturated with 95% O$_2$ and 5% CO$_2$, containing (in mM): N-methyl-D-glucamine (293), KCl (2.5), NaH$_2$PO$_4$ (1.2), HEPES (20), MgCl$_2$ (10), CaCl$_2$ (0.5), NaHCO$_3$ (30), D-glucose (20), thiourea (2), sodium pyruvate (3), N-Acetyl cysteine (12), pH 7.4 (adjusted with HCl). Slices were transferred at room temperature into oxygenated aCSF containing (in mM): NaCl (126), KCl (2.5), NaH$_2$PO$_4$ (1.2), MgCl$_2$ (1.2), CaCl$_2$ (2.4), NaHCO$_3$ (25), D-glucose (11), kynurenic acid (0.25), sodium pyruvate (1), until further processing. Individual slices were then transferred to a recording chamber continuously perfused with recording aCSF heated to 32–34 °C containing (in mM): NaCl (126), KCl (2.5), NaH$_2$PO$_4$ (1.2), MgCl$_2$ (1.2), CaCl$_2$ (2.4), NaHCO$_3$ (25), D-glucose (11), pH 7.4 at 32–34 °C.

GFP+, Td-Tomato+, or GFP+/Td-Tomato+ astrocyte subtypes were visualized in coronal PFC slices under a Nikon FN1 microscope with the appropriate filters. To limit potential differences between cortical layers, astrocytes were consistently recorded in deep cortical layers (layer V-VI), and astrocytes belonging to the three subpopulations were recorded along the same layer, within each slice, for each mouse. Whole-cell patch-clamp recordings were then obtained from these fluorescent astrocytes using a Multiclamp 700B amplifier (Molecular Devices). Recording electrodes were fabricated from borosilicate glass capillaries (1.5 mm OD, 1.12 mm ID; World Precision Instruments), pulled on a Sutter P-97 puller (Sutter Instruments

Company) and had resistances of 6–8 MΩ. These electrodes were filled with an intracellular solution containing (in mM): K⁺-gluconate (105), NaCl (10), KCl (20), $MgCl_2$ (0.15), HEPES (10), EGTA (0.5), ATP (4), GTP (0.3), pH 7.3. The recordings were controlled by PClamp 10 software and digitized with a Digidata 1550B (Molecular Devices), with signals filtered at 10 kHz. Pipette and capacitive currents were neutralized, and after breakthrough, access resistance was compensated.

Electrophysiological data were analyzed offline using Clampfit 10.7 software (Molecular Devices). All recorded cells exhibited a hyperpolarized resting membrane potential (<−74 mV) and a low input resistance (<60 MΩ), which are characteristic of astrocytic electrical properties. Any astrocyte showing more than a 20% change in access resistance was excluded. Astrocyte passive membrane characteristics were assessed by measuring the maximum voltage deflections caused by small current pulses from the holding potential, ensuring that voltage-sensitive currents remained inactive. The input resistance was calculated from the slope of the linear fit to voltage responses evoked by small positive and negative current injections. Astrocytes were then voltage-clamped at their resting membrane potential and subjected to a series of voltage steps ranging from −140 to 0 mV in 10 mV increments. The amplitude of the macroscopic current was determined by subtracting the baseline level (prior to the voltage step) from the average amplitude measured over a 10 ms window beginning 10 ms after the onset of the hyperpolarizing step.

## Behavioral testing
Mice were randomly allocated to their group before LV injection and the experimenter was blinded to the group for all behavioral tests and primary analysis. Mice were handled daily for 2 min for five days before behavioral testing. Then, each mouse was tested on the elevated plus maze, spontaneous alternation test on the Y maze and three-chamber test in a randomized and blinded fashion, over nine days with at least one day off between two tests. Devices were cleaned with 10% ethanol between mice. WT mice were only tested on the elevated plus maze and the three-chamber test as there was no difference in Y maze performance in APP/PS1 mice.

## Elevated plus maze
The elevated plus maze consists of a plus-shaped apparatus with two sets of closed arms perpendicular to two sets of open arms, placed 50 cm above the ground. The maze was illuminated at 150 lux on the open arm extremities, 100 lux on the center, and 50 lux on the closed arm extremities. Each mouse was initially placed at the center of the maze facing an open arm and allowed to freely explore under videotracking for 6 min. Automatic detection with the EthoVision v17 software was performed to extract the distance traveled and time spent in each arm.

## Spontaneous alternation test
The spontaneous alternation test was performed on a Y-shaped maze made of three white plastic arms, each illuminated at 100 lux, placed within an open field arena with visual cues at the end of each arm. Each mouse was placed at the end of the same arm for a 6-min session under video tracking. The total distance traveled and the number of successful alternations (i.e., successive visits of the three different arms) were computed with EthoVision v17.

## Three-chamber test
The three-chamber test was performed in a Plexiglas box (50 × 25 cm) divided into three chambers with two walls each containing a 5-cm door to allow mice to move between compartments. During the first phase (habituation), two empty barred cages were placed in the left and right chambers and the test mouse was left to explore the whole device for 10 min, starting from the central chamber. In the second

phase (social preference), an 8-week-old C57BL/6 J juvenile male mouse was randomly placed in one of the two barred cages, to avoid chamber preference. A Falcon tube filled with colored Lego pieces was placed in the other chamber to serve as the object stimulus. The doors were opened and the test mouse allowed to explore for 10 min. In the third phase (social memory), a novel C57BL/6 J juvenile male was placed in the barred cage that previously contained the object, and the test mouse was left to explore the familiar and novel mouse for 10 min. The test mouse was guided to the central chamber and kept inside with closed doors between two phases. All stages were videotracked with EthoVision v17. The total distance traveled was automatically extracted and the time spent exploring each stimulus was scored manually.

Mice were then euthanized by cervical dislocation at 11-month-old, their brains postfixed for 24 h in 4% PFA, cryoprotected, cut on microtome and used for immunohistological staining as described in the corresponding paragraph.

## Cell cultures
HEK293T cells were maintained in Dulbecco's Modified Eagle Medium F12 (DMEM, #11554546, Fisher Scientific) supplemented with 10% fetal bovine serum (FBS, #17553595, Gibco) and 1% Penicillin-Streptomycin (#11548876, Fisher Scientific). Frozen cells were thawed in a 37 °C water bath and transferred into pre-warmed medium, followed by centrifugation at 330 × $g$ for 3 min. Cell pellets were resuspended in fresh medium, plated and incubated at 37 °C with 5% $CO_2$. For routine passaging, cells were rinsed with 1X D-PBS, detached with 0.05% trypsin (#11580626, Gibco), and split into new flasks with pre-warmed medium. Basal reporter protein expression was very low in HEK cells in these conditions, so we stimulated the STAT3 or the NF-kB pathway with IL6 or TNFα respectively, and performed immunostaining against GFP to amplify CFP or GFP and measure a reliable signal.

For experimental assays, cells were seeded in 24-well plates with a glass coverslip and allowed to reach 65–75% confluence. Cells were then infected with LV-$P_{STAT3}$-GFP or LV-$P_{NF-kB}$-CFPnuc (400 ng of p24 protein per well) in DMEM without FBS for 24 h. The medium was then replaced with fresh culture medium for an additional 24 h recovery period. Cells infected with LV-$P_{STAT3}$-GFP were then incubated for 24 h with IL-6 (#I9646-5UG10, Sigma, 10 ng/ml) or IL-6 + Stattic (#HY-13818, MedChemExpress, 20 µM). Cells infected with LV-$P_{NF-kB}$-CFPnuc were incubated for 24 h with TNFα (#RMTNFAI, ThermoFisher, 10 ng/ml) or TNFα + BAY 11-7082 (#HY-13453, MedChemExpress, 5 µM). DMSO (0.17%) was used as a vehicle control.

Cells were washed with PBS and fixed with 4% PFA for 10 min at room temperature. After three additional washes in PBS, cells were put in blocking solution (PBS with 1% NHS and 0.1% Triton X-100) for 1 h at room temperature. Cells were then incubated with a primary antibody against GFP (Chicken, 1:500, Aves Labs, #GFP-1020) for 48 h at 4 °C. Cells were washed in PBS and incubated with a secondary Alexa Fluor 488 anti-Chicken antibody (1:1,000, Invitrogen) for 1 h at room temperature in the dark. Cells attached to coverslips were finally mounted on a slide with mounting medium. Fluorescence images were acquired using an epifluorescence microscope (Leica DM6000B) with a 20× objective. Three to four fields of view with visible attached and healthy cells on brightfield images were acquired per well, with two wells per condition, from two independent cultures. To isolate specific GFP signal of the LV-$P_{STAT3}$-GFP, an intensity threshold was set relatively to a condition without LV reporter infection, and maintained across all experimental conditions.

## Statistics
Results are expressed as mean ± standard error of the mean. Statistical analyses were performed with GraphPad Prism 10 and R (R Studio v2024.12.1). For each analysis, normality of variables or residuals were

checked by Shapiro-Wilk and visual inspection of residual histograms. Homoscedasticity was assessed by Levene, Spearman or Brown-Forsythe's tests. When necessary, log, square-root or arcsinus transformation of data was applied, as detailed in figure legends. Statistical analyses include linear mixed model (with mouse as a random factor)[72], one- or two-way ANOVA followed by Tukey's post-hoc tests, paired and unpaired $t$ tests (all these tests were two-sided) as well as simple regression, as detailed in each figure legend. The number of astrocytes (n) and mice (N) analyzed per group is provided, but only the average value per mouse is shown. Lines connecting 2 or 3 dots on graphs represent the measures taken in the 2 hemispheres or the 3 subpopulations within the same mouse. For cell cultures, the number of fields of view (n) and independent cultures (N) is provided. The significance level was set at $p < 0.05$. Source Data for all figures and Supplementary Figs. are provided as a Source Data Excel file. Displayed micrographs are representative of different mice per group and cohorts as indicated.

### Reporting summary

Further information on research design is available in the Nature Portfolio Reporting Summary linked to this article.

## Data availability

RNA-seq dataset is available on Gene Expression Omnibus under reference GSE290101. All Source data are provided as a Source Data file. Raw data or images are available from the corresponding author upon request. Source data are provided with this paper.

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

## Acknowledgements

We thank Dr. Caroline Jan for help with confocal analysis, as well as Dr. Karine Cambon, Sueva Bernier, Marjorie Benfissa and Julien Mitja for mouse colony management and behavioral analysis. Part of the image acquisition was done on the NeuroPICT platform at NeuroPSI. We are grateful to Dr. Jan Baijer for initial discussions on FACS analysis, Prof. Takashi Saito, Prof. Takaomi Saido, and Dr Hélène Hirbec for sharing APP$^{NL-F/NL-F}$ mice, Dr. Gilles Bonvento for sharing the PHGDH antibody. We thank Dr. Isabelle Arnoux for help with acute slice experiments, Sydney Barthelemot with post-mortem analysis and Ciana Xu with statistical analysis. This work was supported by CNRS, CEA, INSERM, the Major Research Program of PSL Research University "PSL-Neuro" launched by PSL Research University and implemented by ANR (ANR-10-IDEX-0001 to N.R.), and grants from France Alzheimer (#6173; to N.R. and C.E.), ANR (ANR-16-TERC-0016-01 and ANR21-CE17-0047-02 to C.E.), and FRM (Equipe FRM n°EQU202303016285 to C.E.).

## Author contributions

C.E.: conception; Y.P.B.G., O.G., E.D., R.B., N.R., E.B., and C.E.: experimental design; Y.P.B.G., O.G., V.L., T.B., M.M.C., E.D., C.D., L.J., M.A.C.S., T.L., P.G., F.P., M.G., G.A., N.D., G.M., and C.E.: experimentation & data acquisition; Y.P.B.G., O.G., V.L., T.B., M.M.C., E.D., M.R.P., M.A.C.S., L.B.H., L.S., R.B., N.R., S.B., K.M., E.B., C.E.: data analysis & interpretation; K.B., L.B., N.D., M.C.G., and A.P.B.: reagent generation or supply; R.B.,

N.R., E.B., and C.E.: funding; Y.P.B.G., E.D., and C.E.: figure preparation; C.E.: manuscript writing. All authors revised and approved the manuscript.

## Competing interests

The authors declare no competing interests.

## Additional information

[1]Université Paris-Saclay, CNRS, NeuroPSI, Saclay, France. [2]Université Paris-Saclay, CEA, CNRS, MIRCen, Laboratoire des Maladies Neurodégénératives, Fontenay-aux-Roses, France. [3]Aix Marseille Université, CNRS, Institut de Neurosciences de la Timone (INT), Marseille, France. [4]Université Paris-Saclay, CEA, Centre National de Recherche en Génomique Humaine (CNRGH), Evry, France. [5]Université Paris Cité, Inserm, CEA, Stabilité Génétique Cellules Souches et Radiations, iRCM/IBFJ, Fontenay-aux-Roses, France. [6]Université Paris-Saclay, Inserm, CEA, Stabilité Génétique Cellules Souches et Radiations,iRCM/IBFJ, Fontenay-aux-Roses, France. [7]Collège de France, CNRS, INSERM, PSL-NEURO, Université PSL, Paris, France. [8]These authors contributed equally: Océane Guillemaud, Elisa Degl'Innocenti. [9]These authors contributed equally: Vivien Letenneur, Karouna Bascarane, Tony Barbay, Mie Møller Clausen, Céline Derbois, Martine Guillermier. [10]These authors contributed equally: Solène Brohard, Kevin Muret. ✉e-mail: carole.escartin@cnrs.fr

