## [Transparent Peer Review file · Nature Communications]

Signaling cascades shape functional subpopulations of cortical astrocytes in male wild-type mice and APP/PS1dE9 Alzheimer's disease model

Corresponding Author: Dr Carole Escartin

Version 0:

Reviewer comments:

Reviewer #1

(Remarks to the Author)

This is an interesting study that develops novel viral-based tools to identify astrocyte states *in vivo*, based on induction of STAT3 or NFκB signaling pathways coupled to a fluorescent reporter. The authors use these tools to investigate if the proportion of astrocytes that are active for these signaling pathways is impacted by amyloid pathology, using an Alzheimer's disease mouse model. They further investigate if there are functional differences in these cells using a number of assays. These reporters have the potential to be powerful tools for future studies, however there are some important control experiments (outlined below) that need to be included, in order to verify that the reporters are reflecting true astrocyte states *in vivo*.

1. The study relies on local injection of lentivirus to drive expression of the reporters and to define astrocyte sub-states. Important controls are needed to verify these experiments. First, can the authors perform immunostaining in uninjected mice for these active pathways e.g. pSTAT3, and confirm that these astrocyte states exist in the same proportions in WT/AD mice in the absence of damage induced by local injection or induction by lentivirus infection. Second, can similar staining be performed in injected mice to verify that the reporter fluorescence faithfully reports the endogenous signaling pathway.
2. In Figure 3, it is concluded that the level of STAT3 or NFκB pathway activity is unaffected by proximity to amyloid plaques, based on fluorescence. To make this conclusion experiments need to be performed to identify the dynamic range of the reporter e.g. is the production of fluorescent protein already saturated (the authors demonstrate that fluorescence can be decreased by a pathway antagonist)?
3. For the RNA sequencing, a number of the groups only contain two biological replicates. The rigor of the quality control is appreciated that led to removal of these samples (from an original 5 per group, explained in the methods), however it calls into question if there are enough replicates to be performing statistical comparisons and making strong conclusions about gene expression/pathway differences between the astrocyte groups. Is it possible to combine samples from the WT and APP groups to increase the power of the analysis, or do the authors think these are too distinct already based on the pathology?
4. Related to the quality of the RNA sequencing data, it seems surprising that connexin genes are not detected in the NFκB astrocytes in WT mice (Fig S2), whereas they are in AD mice (Fig 6). Is this a true genotype-driven difference, or technical? It does not seem to correlate with the finding that NFκB astrocytes do show functional gap junction coupling/hemichannel activity.
5. Discussion line 659, about hemichannel activity in WT and AD mice. Were these groups compared statistically in order to make the conclusion that there are differences across genotypes?
6. It would be beneficial to provide the list of DEGs from the RNA sequencing as a supplemental table.
7. In Figure 7, the authors investigate if inhibiting the STAT3 or NFκB pathway in astrocytes impacts behavior in AD mice. There are effects in some behaviors, but these are not necessarily related to AD deficits. Therefore, manipulating these pathways in astrocytes does not strongly improve AD behavior (although see next point), and in some cases makes it worse

e.g. in social memory in the 3-chamber test. As there is no AD deficit to begin with in the social testing, the authors need to perform the same experiments in WT mice if they want to make the conclusion that these differences are specifically related to AD (as mentioned in the Discussion paragraph line 669), i.e. are these pathways beneficial for general astrocyte function even in a healthy context?

8. There is a behavioral alteration in AD mice vs WT in the elevated plus maze, which is ameliorated with NFKB blockade. However, statistical comparisons are only performed against the WT group. The authors should provide the comparisons for all groups, or use the AD group as the comparator if they use a one-way ANOVA.

9. The manuscript moves back and forth between studies in WT mice and AD models, and it's not always clear when these transitions occur. The model being studied should be made clearer at each point.

Reviewer #2

(Remarks to the Author)

Signaling cascades shape functional subpopulations of cortical astrocytes in wild-type and Alzheimer's disease mice.

Giraudon et al.

In this study, Giraudon and colleagues use astrocyte-specific reporters for the STAT3 and NF- κ B signalling pathways to identify at least three astrocyte subpopulations in the prefrontal cortex (PFC) of an Alzheimer's disease (AD) mouse model. Notably, these signalling-defined subpopulations are not induced by amyloid deposition and are also present in wild-type mice. These subpopulations exhibit distinct morphologies, molecular signatures, and functional profiles. NF- κ B⁺ astrocytes occupy larger territories and show increased lysosomal activity, whereas STAT3⁺ astrocytes display enhanced connexin hemichannel opening. Selective inhibition of these subpopulations in AD mice reduces amyloid plaque size in the prefrontal cortex and modulates behaviours such as anxiety, social preference, and memory.

Overall, this study provides valuable insights into distinct astrocyte subpopulations, defined by STAT3 and NF- κ B signalling, in the mouse brain. Unfortunately, in my opinion, the claims relating to astrocyte subpopulations and function in the wildtype brain are weak at present and require some more work, or textual clarifications, to fully support the claims made. On the other hand, the manuscript does report interesting findings on how these subtypes impact AD/amyloid-type pathology. These appear to be much more solid and offer a promising framework for understanding astrocyte roles in neurodegeneration.

Major Issues:

1) The study makes use of clever vector systems, which express fluorescent reporters when either the STAT3 or NF- κ B signalling pathways are active in astrocytes. The authors report that these are expressed in astrocytes in the wildtype brain. However, I do not share the authors' view that normal neuronal density, together with normal microglial numbers and morphology, necessarily reflect lack of reactivity induced by vector transduction. This is especially the case because the authors use GFAP as an astrocyte marker, which is generally thought to be low in prefrontal cortex, increasing on injury and/or disease. Or are the vectors exhibiting a bias in cell transduction? Do the authors have data reporting LV system performance measured using other astrocyte markers? Can the authors provide independent evidence that the reporters faithfully recapitulate the wildtype situation using published single cell data, or ideally spatial transcriptomics data?

2) Following on from (1), while I understand the rationale to compare the STAT3⁺ and NF- κ B⁺ populations, I think full sequencing data for all three populations (including the dual STAT3⁺/NF- κ B⁺ population) in wildtype and AD conditions should be provided. Given their distinct effects on astrocytes, it would be interesting to investigate whether one pathway antagonises or regulates the other. Experiments involving combined inhibition of both pathways could reveal whether modulating one affects the activation of the other. Further investigation into how the STAT3 and NF- κ B pathways might influence each other, either through direct molecular interactions or by modulating shared downstream targets, would be valuable to better understand their functional relationship (although this may be for follow up papers).

3) Personally, I found the manuscript difficult to follow with the constant jumps between wildtype and AD conditions. Would it not be more logical to discuss the wildtype situation comprehensively and then move to AD? As it stands, the emphasis of the paper seems focused on AD - is this what is intended? Furthermore, what the authors are really dealing with is a model of Alzheimer's type amyloidosis (APP/PS1) – not Alzheimer's disease *per se*. The manuscript would also benefit from a description of the timeline for plaque deposition and the appearance of behavioral deficits, which was not readily apparent. Finally, if the emphasis really is on the AD/amyloid context, are the effects seen in the APP/PS1 mouse line mirrored in other mouse lines, which would be expected if the results are broadly applicable to AD-type amyloidosis.

4) I agree with the authors that many of the studies describing astrocyte subtypes have not adequately addressed functionality. In my opinion, whether the authors actually address this issue is, however, debateable. Certainly, I feel that the FRAP measurements of gap junction coupling, hemichannel activity assays and plaque measures provide little extra information than, for example, differential Ca²⁺ signaling in astrocyte subtypes (for example PMIDs: 32139688 and 18575586).

The authors did describe differences in several behavioral tests run following the inhibition of STAT3 or NF- κ B+ signalling. However, these tests were performed largely in the AD/amyloid context and do not provide information (in my mind) on the effects of pathway inhibition in wildtype mice. Hence, I do not think the observed behavioural changes necessarily can be extrapolated to functional importance in the wildtype situation.

Finally, did the authors attempt to block both pathways simultaneously and look at the behavioral outcomes (even in the AD/amyloid context)? What were the effects?

Minor issues:

1) The quality of the figures in the manuscript provided for review was low. Higher-resolution images and clearer figure presentation are needed to ensure data interpretation and reproducibility. Furthermore, in Figure 2 and Supplemental Figure 1, it would be beneficial if the authors avoided the combination of red and green to ensure figures are accessible to color-blind readers.

2) In general, the statistical analyses used are well described in the 'Methods' section and appear appropriate. Significance is defined as $p < 0.05$. However, in the figures it appears multiple p-values are considered (multiple *), but the significance levels are (generally) not given on the figure itself, or in the legend.

3) Full details on the systems and parameters used to acquire fluorescence images should be provided in the 'Methods', rather than merely stating 'Images were acquired with an inverted scanning confocal microscope'. How were thresholds for image analysis set?

4) The use of GFAP staining to report morphology has, to the best of my knowledge, largely been discounted in favour of the use of membrane-localized fluorescent reporters, which more accurately report morphology. At the least, the caveats of GFAP use should be discussed.

5) In my opinion, the manuscript would be helped by explanations of how the cathepsin and proteasome probes work.

6) In line 108, only Fig. 1a is referenced, whereas the text description appears to relate to more than just Fig. 1a.

7) In line 493, the figure number associated with the text is missing and should be added.

8) In lines 555 and 562, the phrase "data not shown" should be avoided. Please include the relevant data, even as supplementary material, to substantiate the claims made.

9) In Fig. 1c, the authors present the quantification of the total percentage of fluorescent cells co-expressing different cell-specific markers; however, they do not provide the values for each marker quantification. From figure, it seems odd that the percentage of astrocytes expressing reporters is so low (approx 70%) if specificity is high, especially considering the low amount of off-target expression detected in the multiple cell types assayed (< 5%). Why do the combined values not add up to 100%?

10) In Fig. 5f and 5g it would be interesting to also show the data regarding the population expressing both reporters.

Reviewer #3

(Remarks to the Author)

Poulot-Becq Giraudon and colleagues describe three molecular clusters of astrocytes based on their Nf κ B and STAT3 transcription factor activity using novel lentiviral fluorescent reporter probes in WT and APP/PS1 mice co-injected in the mouse prefrontal cortex. They nicely validated the co-injection/expression approach by co-injecting two lentiviral vectors encoding the expression of either GFP or Td-tomato driven by an ubiquitous astrocytic promoter (PGK) followed by FACS and showing a remarkable co-expression in the same astrocytes. Surprisingly, they show that all three subpopulations of astrocytes exist in similar proportions across WT and APP/PS1 mice and that their proportions are not impacted by proximity to amyloid plaques in APP/PS1 mice, suggesting that these three subpopulations of astrocytes are developmentally different by as-yet-unknown epigenetic mechanisms. They show morphological (via Sholl analysis), transcriptomic (via bulk RNA-seq), electrophysiological (via patch-clamp), and hemichannel activity (using ethidium bromide loading) differences between Nf κ B+ and STAT3+ astrocytes. Nf κ B+ astrocytes are larger, have more lysosomal activity and perhaps less proteasomal activity, and less hemichannel activity than STAT3+ astrocytes. Inhibition of STAT3 with LV-SOCS3 and of Nf κ B with LV-NFKBIA reduced plaque size but altered social memory in the 3-chamber test.

Overall, this is a well-written and very interesting paper that not only adds more evidence to the recent literature on the molecular heterogeneity of astrocytes in the normal and AD brain but also provides both innovative tools to study the molecular pathways driving reactive astrogliosis and new insights into the downstream consequences of the activity of Nf κ B and STAT3, two of the main transcription factors involved in reactive astrogliosis in AD. The characterization of astrocytes in WT and APP/PS1 mice with morphological, electrophysiological, transcriptomics, behavioral, and functional ex-vivo assays is commendable. The notion that these different molecular subpopulations of astrocytes are epigenetically determined in the normal brain is quite intriguing and novel.

I have several constructive questions, comments and suggestions:

1. The choice of prefrontal cortex over hippocampus is not well explained. Certainly, the choice of behavioral tests is adequate to evaluate the function of the prefrontal cortex but is there any particular reason to avoid hippocampus for this approach given its importance in AD?
2. I am a bit surprised that proximity to plaques did not influence reporter expression. Have prior studies shown p65 and pSTAT3 immunoreactivity in reactive astrocytes around plaques? The fact that SOCS3 and NFKBIA lentiviral expression reduced plaque size also argues that STAT3 and NfκB are activated in astrocytes around plaques. I have several suggestions to reanalyze and/or clarify these analyses:
 - a. It may be unfair to include NfκB+ astrocytes in the GFP fluorescence vs. distance to plaque regression in Fig 3b and STAT3+ astrocytes in the CFPnuc fluorescence vs. distance to plaques in Fig. 3c since they were classified as NfκB+ or STAT3+ based on their low fluorescence intensity for GFP and CFPnuc, respectively. If the regression is repeated excluding the “negative” group there may be a statistically significant negative correlation between GFP or CFPnuc fluorescence and distance to the nearest plaque.
 - b. If the denominator of the proportions in Fig. 3d is the total number of astrocytes of each cluster, isn't the fact that the proportion of the three clusters of astrocytes is highest in the 41-80 microns interval evidence that there is an effect of plaque proximity?
 - c. The distance to the nearest plaque was measured as the distance between the center of the plaque and the center of the astrocyte cell body. Therefore, this distance includes the plaque diameter, which can be anywhere from 5 to 100 microns depending on the size of the plaque. Astrocyte cell bodies are not embedded in the amyloid plaque (unlike microglia cell bodies, which may be) but they are pushed away by the plaque mass effect (see repulsion effect described by Galea E et al. PNAS 2015 in the same mouse model). This explains why there are few astrocytes in the first bins and why the proportion for all three clusters is lower in the 0-40 microns bin. Measuring distance from the edge of the nearest plaque would have prevented this issue and likely shown an effect of plaque proximity.
 - d. Can these results be validated with Aβeta, p65 and pSTAT3 multiplex immunohistochemistry?
 - e. If the conclusion is that plaque proximity did not make a difference in reporter fluorescence, could the reason be that reactive astrocytes around amyloid plaques were less amenable to lentiviral transduction due to an antiviral interferon-mediated response to plaques as shown by the Liddelow lab? Please, comment on this possibility.
3. Were the morphological and ephys analyses of the three astrocyte subpopulations controlled for cortical layer? In other words, could the differences noted across molecular subtypes be due to differences in the layer location when selecting the astrocytes to be analyzed? While the authors did not find differences in subtype proportions across layers, a layer-by-layer comparison of these morphological parameters could be helpful to ensure that the cortical layer where the astrocyte sits is not driving the differences observed.
4. RNA-seq data:
 - a. Were pathway enrichment analyses in Table 1 done with a rank-based gene set enrichment analysis or using a hypergeometric overrepresentation test? Is the p-value in Table 1 adjusted for multiple hypothesis testing (e.g. FDR)? Should the pathway enrichment analyses in Table 1 be done separately for the 477 up and the 158 down DEGs in STAT3+ vs. NfκB+ astrocytes? What are the pathways enriched in NfκB+ vs. STAT3+ astrocytes in both APP/PS1 and WT mice? Also, the KEGG database is very biased towards infectious pathogens. Could the authors try GO-BP and Reactome databases for their pathway enrichment analyses?
 - b. Transcription factors upstream DEGs were evaluated using Pscan, which is based on the presence of known binding motif sequences, however this sequence-based approach could yield non-specific transcription factors. A Transcription Factor Enrichment Analysis based on the ENCODE ChIP-Seq database would be helpful to complement this analysis.
 - c. Fig. 5c: Do astrocytes express Ig Fc receptors, Clec7a, Cd68? Aren't these microglial genes? Could the authors provide a supplementary Excel spreadsheet with the differential expression analysis results at the gene level?
5. Can the authors speculate why SOCS3 expression only lowered GFP fluorescence by 26% after 3 months? Could this be due to longer half-life of GFP vs. CFP? Why the authors think this is enough to trigger changes in plaque size and behavior?
6. Statistics: were astrocytes from the same mouse treated as independent observations in one-way ANOVA and 2-way ANOVA or did the authors use the average per mouse in the ANOVA? The former would imply pseudoreplication and mixed effect models are recommended to avoid it.

Minor:

1. Lines 186-187. The authors state “The PFC of three mice were pooled, resulting in N=5 independent PFC samples from APP mice.” However, Figure 2d shows: Proportion of each astrocyte subpopulation, obtained from three independent cohorts with each N=6-7 APP mice. Can the authors reconcile these apparently discrepant statements? If the cell sorting experiment was repeated 3 times, can they show an additional stack bar plot with average + s.d. or s.e.m. error bars in Figure 2d?
2. Throughout the manuscript, please consider referring to APP mice as APP/PS1 mice.
3. Suppl. Fig. 3: I believe the authors mean singlets rather than singulets.
4. Suppl. Fig. 4b: Please, consider labeling panel as “photobleach” rather than “bleach.”

Reviewer #4

(Remarks to the Author)

Version 1:

Reviewer comments:

Reviewer #1

(Remarks to the Author)

The authors have responded to all of the points raised in my first review with new experiments, rearranging of existing data, and by adding clarifying statements to the text. I have no further concerns.

Three minor points:

1. Given Reviewer 3's query about 'microglia' genes being detected in the sequencing data, a suggestion is to also include expression level data in the supplementary RNAseq table to give an indication of how highly these genes are expressed in the astrocyte samples.
2. Figure 5D, STAT3/NFKB column for the APP sample - there is an extra datapoint in the column (3 instead of 2)
3. The GEO record is set to private so could not be accessed

Reviewer #2

(Remarks to the Author)

Signaling cascades shape functional subpopulations of cortical astrocytes in wild-type and Alzheimer's disease mice.

Giraudon et al.

In this study, Giraudon and colleagues use astrocyte-specific reporters for the STAT3 and NF- κ B signalling pathways to identify at least three astrocyte subpopulations in the prefrontal cortex (PFC) of an Alzheimer's disease (AD) mouse model. Notably, these signaling-defined subpopulations are not induced by amyloid deposition and are also present in wild-type mice. These subpopulations exhibit distinct morphologies, molecular signatures, and functional profiles. NF- κ B⁺ astrocytes occupy larger territories and show increased lysosomal activity, whereas STAT3⁺ astrocytes display enhanced connexin hemichannel opening. Selective inhibition of these subpopulations in AD mice reduces amyloid plaque size in the prefrontal cortex and modulates behaviours such as anxiety, social preference, and memory. Hence, the study demonstrates that intrinsic signaling activity defines astrocyte subpopulations in the mouse PFC and highlights that these subpopulations exert distinct and complex effects on hallmark AD alterations.

This is a resubmission of a previously reviewed manuscript.

Overall, this is a well written and interesting paper that adds considerable weight to our understanding of astrocyte (molecular) heterogeneity in both the healthy and diseased brain. It is obvious the authors made substantial efforts during the revision and, as such, the manuscript is very much improved. However, I would like clarification regarding experimental data added to the manuscript and raise a number of minor comments for consideration by the authors.

Major Issue:

i) The authors added new behavioral data on wild-type animals which shows no effects on anxiety, social preference and memory in these mice following pathway inhibition. This is in contrast to the effects seen in APP/PS1 mice. From the manuscript, it appears that the wild-type testing was done during the revision period (independent cohort). However, I am not sure that the data obtained and comparisons made are as reliable as implied (e.g. Fig 7c-f, Supp Fig 5 a-d) - as sensitivity to the different testing conditions that were undoubtedly present may impact animal performance in an unpredictable way.

Minor Issues:

i) In my opinion, there are instances of hyperbole which can (and should be) toned down, without the paper losing any of its impact. For example, the authors claim 'results show that PFC astrocytes display signaling heterogeneity in WT mice and throughout disease progression in APP/PS1 mice' when the latest timepoint they assess in the mice is 12 months of age. APP/PS1 mice have a much longer lifespan (Roberts et al. doi: 10.1101/2024.10.15.618508).

Likewise, I am still of the opinion that the authors provide no direct proof of functional differences in the text (original review point 4). What they actually report is differences in proteasome activity using reporter assays, SR101 diffusion between gap junctions and ethidium bromide uptake via hemichannels. While this is highly suggestive of functional alterations, there is no direct evidence that the functional output of the cell is changing. Likewise, while the authors show an effect of astrocyte manipulation on behavior, it is unclear exactly what the underlying mechanistic action driving this effect is. The papers cited in review (Batiuk et al. and Takata and Hirase) reported distinct physiologies (Ca²⁺ dynamics) of cells without a clear functional output.

ii) On reflection, I agree with Reviewer 1's comment on the number of samples used for RNA-seq. I do not necessarily disagree with the author's response, especially given the data they show in the rebuttal, but I do think that some reference to the low sample number (and the limits of interpretation) should be made.

iii) I found it hard to interpret the DEG numbers in Fig. 5A and B: can the authors please clarify?

iv) The inhibition of the NF- κ B pathway in wild-type mice has an effect on the total distance covered in the 3-chamber test, but, as far as I can see, any discussion of this is omitted from the manuscript. Why? How should it be interpreted?

Reviewer #3

(Remarks to the Author)

The authors have satisfactorily answered all my questions and comments. I appreciate the additional bioinformatic analyses, the reanalysis of distance to plaque, and the application of mixed effect models. As per requests from the other reviewers, the authors have added in vitro validation experiments in HEK cells and replicated the three subpopulations of astrocytes (STAT3+, NF- κ B+ and STAT3+/NF- κ B+) in the APP-NL-F/NL-F mouse model.

Although I did not ask for the in vitro validation experiments of the probes, they seem to confirm a point I raised, that the STAT3-GFP probe seems to be less responsive to known STAT3 inhibitors (Stat3i, SOCS3) than the NF- κ B probe to known NF- κ B inhibitors (BAY, NFKBIA) as judged by the differences in effect size in Figures 1d vs. 1e and 1f vs. 1g. Could the authors comment whether these differences are more likely to be technical or biological? What are the implications of such difference for the interpretation of the work? Why not using primary astrocytes or an astrogloma cell line?

Minor points:

1. For the readership, it'd be great if the overlapping genes in Table 3 are expressed as gene symbols in addition to Ensembl ids.
2. Introduction and Material and Methods - Animals: please add the Greek letter beta before amyloidosis.
3. What do the white and black circles mean in new Figures 1d-e, cultures #1 and #2? Please, explain in the legend.

Reviewer #4

(Remarks to the Author)

Version 2:

Reviewer comments:

Reviewer #2

(Remarks to the Author)

The authors have answered all my queries and I congratulate them on a very interesting manuscript.

Reviewer #4

(Remarks to the Author)

We thank the reviewers for their positive assessment of our work, constructive comments, and helpful suggestions. We provide a point-point reply below. We have extensively revised and complemented the manuscript following reviewer's suggestions. All editions made to the text in response to the reviewers appear in blue.

Specifically, in response to comments by Reviewer 1 and 2, we have combined data obtained in WT mice with those of APP/PS1 mice (new **Fig 5b, 5d, 5e, 6a, 6c, Supplemental Fig. 3b, 3d**). The differences between astrocyte subpopulations and the effect of the genotype can now directly be visualized and assessed statistically. In addition, a new set of behavioral analysis was performed on WT mice to complement **Fig. 7** (new **Supplemental Fig. 5e-g**), showing that subpopulation targeting impacts anxiety, social preference and memory only in APP/PS1 mice. We also show that these subpopulations are also observed in another amyloidosis model (**Supplemental Fig. 1b**). We also provide STAT3 immunofluorescence quantification in WT, uninjected mice, confirming that PFC astrocytes display variable levels of this transcription factor, suggesting different ability to signal through this cascade (**Supplemental Fig. 1g**). Finally, we provide new *in vitro* data using pharmacological inhibitors of the two pathways, confirming that the two reporters are sensitive to pathway activity (**Fig. 1d, e**). On a separate note, based on a comment received at a conference, we have also calculated the speed of recovery during FRAP experiments to confirm that gap-junction coupling is not different among subpopulations (**Supplemental Fig. 4c, 4f**).

We hope that the reviewers and editor will appreciate the thorough edition of the manuscript and its significant improvement over the previous version, strongly supporting our original findings.

Reviewer #1 (Remarks to the Author):

This is an interesting study that develops novel viral-based tools to identify astrocyte states *in vivo*, based on induction of STAT3 or NF κ B signaling pathways coupled to a fluorescent reporter. The authors use these tools to investigate if the proportion of astrocytes that are active for these signaling pathways is impacted by amyloid pathology, using an Alzheimer's disease mouse model. They further investigate if there are functional differences in these cells using a number of assays. These reporters have the potential to be powerful tools for future studies, however there are some important control experiments (outlined below) that need to be included, in order to verify that the reporters are reflecting true astrocyte states *in vivo*.

We thank the reviewer for the positive feedback and interest in our findings and tools.

1. The study relies on local injection of lentivirus to drive expression of the reporters and to define astrocyte sub-states. Important controls are needed to verify these experiments. First, can the authors perform immunostaining in uninjected mice for these active pathways e.g. pSTAT3, and confirm that these astrocyte states exist in the same proportions in WT/AD mice in the absence of damage induced by local injection or induction by lentivirus infection. Second, can similar staining be performed in injected mice to verify that the reporter fluorescence faithfully reports the endogenous signaling pathway. P-STAT3 staining is notoriously difficult to obtain given the lability of the phosphor-Tyrosine epitope, especially in chronic neurodegenerative models or in WT mice, where STAT3 activation may be low, transient or asynchronous. p-STAT3+ border astrocytes are reported in acute diseases such as spinal cord injuries or around brain tumors (refs 52, 53 in manuscript). The fact that p-STAT3 or p-NF- κ B were undetectable in our brain sections is precisely why we developed these two reporters, to provide a more sensitive *in situ* monitoring of cascade activities. This is now clarified in the introduction: "*The active, phosphorylated forms of STAT3 and NF- κ B are unreliably*

detect in the mouse brain, especially in chronic diseases with mild activation. Therefore, to assess the activity of these two pathways in situ and understand their role in generating reactive astrocyte heterogeneity, we developed two lentivirus-based reporters”, and at the beginning of the Result section: “To efficiently monitor the activity of two central signaling cascades regulating astrocyte changes in AD, we generated two lentivirus (LV)-based reporters, circumventing the low detection of phosphorylated active forms of STAT3 and NF-κB by immunostaining.”

We performed STAT3 immunostaining in uninjected Aldh1L1-eGFP mice to assess STAT3 pool in PFC astrocytes, as a measure of their intrinsic ability to signal through this pathway. We found 65% STAT3+/GFP+ astrocytes cells. It shows that PFC astrocytes have variable levels of STAT3 and thus potential STAT3 activity. Interestingly, this percentage is quite similar to the percentage obtained with our STAT3 reporters in WT mice were ~79% of astrocytes were STAT3 + (either STAT3+ only or STAT3+/NF-κB+, **Fig. 2d**). This data is now provided as **Supplemental Fig. 1g**). The same analysis was attempted with NF-κB, unfortunately none of the tested antibodies provided a reliable staining on PFC mouse sections.

As an additional demonstration that these reporters faithfully report the activity of the pathway, we originally used the two well-described pathway inhibitors (SOCS3 and NFKBIA, now **Fig. 1f, g**), reporting significant reduction in fluorescent reporter protein levels. We now provide new data with pharmacological inhibitors in cell cultures (**Fig. 1d, e**; see next point), further validating that the two reporters are sensitive to their cognate pathway in two different experimental settings.

2. In Figure 3, it is concluded that the level of STAT3 or NFKB pathway activity is unaffected by proximity to amyloid plaques, based on fluorescence. To make this conclusion experiments need to be performed to identify the dynamic range of the reporter e.g. is the production of fluorescent protein already saturated (the authors demonstrate that fluorescence can be decreased by a pathway antagonist)? We did include data as **Fig. 1d** and **1e** in the original manuscript (now **Fig. 1f** and **1g**), showing that SOCS3 and NFKBIA respectively, significantly decreased the fluorescence of their cognate LV reporter. We now added *in vitro* experiments reporting a significant decrease in fluorescence 24h after treatment with pharmacological inhibitors (**Fig. 1d, e**), showing that indeed these reporters respond dynamically to cascade activity.

Regarding saturation, which is indeed a critical point, we remind the reviewer that we observed very variable levels of GFP and CFP fluorescence in astrocytes, meaning that if some astrocytes may express maximum levels of the reporter proteins, most of them express intermediate levels (see **Fig. 3b, c**), that could be further induced by plaques if such a mechanism was at stake. This does not seem to be the case.

In response to Reviewer 3, point 2, we have also performed complementary analyses confirming that amyloid plaques do not increase STAT3 or NF-κB activity. It does not however mean that plaques do not activate other intracellular pathways or change astrocyte transcriptome. Indeed, we now emphasize that even if astrocytes of the three subpopulations are found both in WT and APP/PS1 mice, their molecular profile is significantly different (see also reply to next point).

3. For the RNA sequencing, a number of the groups only contain two biological replicates. The rigor of the quality control is appreciated that led to removal of these samples (from an original 5 per group, explained in the methods), however it calls into question if there are enough replicates to be performing statistical comparisons and making strong conclusions about gene expression/pathway differences between the astrocyte groups. Is it possible to combine samples from the WT and APP groups to increase the power of the analysis, or do the authors think these are too distinct already based on the pathology? We do not wish to combine WT and APP/PS1 samples because we do report several molecular and functional differences between these two genotypes (e.g. *Connexin* mRNA levels (see new **Fig 6a**), which translate into different hemichannel activities (see new **Fig. 6c**). In fact, there are 467 DEG between WT-STAT3+ and APP-STAT3+ astrocytes, showing that astrocyte

subpopulations are indeed impacted by the amyloidosis pathological context. This is now mentioned in the text and DEG are provided in **Table 1**.

We acknowledge the limitation of having only two replicates in some groups due to stringent quality control criteria, but we do find hundreds of DEG with very low p values and high log₂FC (see **Table 1**). Moreover, even after re-integrating the excluded samples, the same pattern of expression across subpopulations was observed for genes of interest (see for example for *Ctz* and *Gjb6* below and in **Fig 5d** and **6a**).

Despite this limitation, we are confident about the reported differences in *cathepsin* and *connexin* gene expression, as they were functionally validated by differences in cathepsin activity and hemichannel activity, respectively.

4. Related to the quality of the RNA sequencing data, it seems surprising that connexin genes are not detected in the NFκB astrocytes in WT mice (Fig S2), whereas they are in AD mice (Fig 6). Is this a true genotype-driven difference, or technical? It does not seem to correlate with the finding that NFκB astrocytes do show functional gap junction coupling/hemichannel activity. Both technical and biological explanations can be considered. We know that due to a low number of sorted cells per population, some low-expressed transcripts may not be detectable, explaining that WT-NF-κB⁺ astrocytes exhibit undetectable levels for both connexins. *Gjba1* and *Gjba6* are better detected in AD-NF-κB⁺ astrocytes than in WT-NF-κB⁺ astrocytes because both genes are known to be induced in AD astrocytes, as reviewed in Koulakoff *et al.* 2012 (DOI: 10.1016/j.bbamem.2011.10.001), including in 12-month-old APP/PS1DE9 mice (Yu *et al.*, Mol Neurobiol, 2025; DOI: 10.1007/s12035-024-04536-3), which is also visible in our own RNASeq data (see new **Fig. 6a** for a direct comparison).

5. Discussion line 659, about hemichannel activity in WT and AD mice. Were these groups compared statistically in order to make the conclusion that there are differences across genotypes? The 6 groups (3 subpopulations in two genotypes) are now directly shown and statistically compared in **Fig. 6e**, using the linear mixed model statistical analysis suggested by reviewer 3. This analysis confirms the previously described differences between subpopulations among APP/PS1 or WT mice but also reports a very significant “genotype x subpopulation” effect, supporting our point in the discussion.

6. It would be beneficial to provide the list of DEGs from the RNA sequencing as a supplemental table. We now provide all DEG lists as **Table 1**: DEG between the three subpopulations among APP/PS1 and WT mice, as well as DEG in the same subpopulation between WT and APP/PS1 mice. Pathway analysis is now shown as **Table 3**.

7. In Figure 7, the authors investigate if inhibiting the STAT3 or NFKB pathway in astrocytes impacts behavior in AD mice. There are effects in some behaviors, but these are not necessarily related to AD deficits. Therefore, manipulating these pathways in astrocytes does not strongly improve AD behavior (although see next point), and in some cases makes it worse e.g. in social memory in the 3-chamber test. As there is no AD deficit to begin with in the social testing, the authors need to perform the same experiments in WT mice if they want to make the conclusion that these differences are specifically related to AD (as mentioned in the Discussion paragraph line 669), i.e. are these pathways beneficial for general astrocyte function even in a healthy context? We thank the reviewer for this suggestion. We performed a new set of behavioral tests on WT mice. We find that subpopulation targeting by SOCS3 or NFKBIA does not impact anxiety, social preference and memory in WT mice (Supplemental Fig. 5e-g). We can thus conclude that astrocyte subpopulations controlled by the STAT3 and NF-κB pathways impact these behaviors only in an AD context.

8. There is a behavioral alteration in AD mice vs WT in the elevated plus maze, which is ameliorated with NFKB blockade. However, statistical comparisons are only performed against the WT group. The authors should provide the comparisons for all groups, or use the AD group as the comparator if they use a one-way ANOVA. We apologize if this point was unclear. We did use a one-way ANOVA to compare EPM performance between all groups. The only significant differences were between WT-CTR and APP-CTR or APP-SOCS3 (with \$p\$ values now indicated in Fig. 7c legend). Tukey tests among the three APP/PS1 groups gives non-significant \$p\$ -values \$p > 0.676\$, which is now also indicated in the legend. We acknowledge that the lack of significant difference between APP/PS1-CTR and APP/PS1-NFKBIA mice mitigates the impact of NFKBIA improving effects, this is why we used the sentence “suggesting slightly improved anxiety levels”, already in the original version.

9. The manuscript moves back and forth between studies in WT mice and AD models, and it's not always clear when these transitions occur. The model being studied should be made clearer at each point. We apologize for this, we have thoroughly edited the manuscript, (also in response to reviewer 2, point 3), and now hope that clarity is improved. In addition, as stated before, WT data are now included in main figures.

Reviewer #2 (Remarks to the Author):

In this study, Giraudon and colleagues use astrocyte-specific reporters for the STAT3 and NF-κB signalling pathways to identify at least three astrocyte subpopulations in the prefrontal cortex (PFC) of an Alzheimer's disease (AD) mouse model. Notably, these signalling-defined subpopulations are not induced by amyloid deposition and are also present in wild-type mice. These subpopulations exhibit distinct morphologies, molecular signatures, and functional profiles. NF-κB⁺ astrocytes occupy larger territories and show increased lysosomal activity, whereas STAT3⁺ astrocytes display enhanced connexin hemichannel opening. Selective inhibition of these subpopulations in AD mice reduces amyloid plaque size in the prefrontal cortex and modulates behaviours such as anxiety, social preference, and memory.

Overall, this study provides valuable insights into distinct astrocyte subpopulations, defined by STAT3 and NF-κB signalling, in the mouse brain. Unfortunately, in my opinion, the claims relating to astrocyte subpopulations and function in the wildtype brain are weak at present and require some more work, or textual clarifications, to fully support the claims made. On the other hand, the manuscript does report interesting findings on how these subtypes impact AD/amyloid-type

pathology. These appear to be much more solid and offer a promising framework for understanding astrocyte roles in neurodegeneration.

We thank the reviewer for the positive feedback and valuable suggestions. We hope that he/she will be convinced by the revised version with additional data on WT mice, including new behavioral analysis.

Major Issues:

1) The study makes use of clever vector systems, which express fluorescent reporters when either the STAT3 or NF- κ B signalling pathways are active in astrocytes. The authors report that these are expressed in astrocytes in the wildtype brain. However, I do not share the authors' view that normal neuronal density, together with normal microglial numbers and morphology, necessarily reflect lack of reactivity induced by vector transduction. This is especially the case because the authors use GFAP as an astrocyte marker, which is generally thought to be low in prefrontal cortex, increasing on injury and/or disease. Or are the vectors exhibiting a bias in cell transduction? Do the authors have data reporting LV system performance measured using other astrocyte markers? Can the authors provide independent evidence that the reporters faithfully recapitulate the wildtype situation using published single cell data, or ideally spatial transcriptomics data? Quantification in **Fig. 1c** was done on APP/PS1 mice, explaining the presence of GFAP+ astrocytes in the PFC. We have now assessed LV reporter tropism by co-immunostaining with the ubiquitous astrocyte marker phosphoglycerate dehydrogenase (PHGDH) in WT and APP/PS1 mice. We found that more than 88% GFP and/or CFP cells co-express the astrocyte marker PHGDH. We also provide the percentage of PHGDH co-expression within each subpopulation in APP/PS1 mice in **Supplemental Fig. 1a**. It shows that both reporters have a strong astrocyte tropism and that all three subpopulations are astrocytes with no bias in transduction.

We also added data on WT mice showing that 65% of PFC astrocytes (identified by their GFP expression in Aldh1L1-eGFP mice) co-express STAT3. This new data shown in **Supplemental Fig. 1g** demonstrate that indeed astrocytes display variable levels of this transcription factor, with numbers matching those obtained with our LV reporter (see reply to reviewer 1, point 1).

2) Following on from (1), while I understand the rationale to compare the STAT3+ and NF- κ B+ populations, I think full sequencing data for all three populations (including the dual STAT3+/NF- κ B+ population) in wildtype and AD conditions should be provided. Given their distinct effects on astrocytes, it would be interesting to investigate whether one pathway antagonises or regulates the other. Experiments involving combined inhibition of both pathways could reveal whether modulating one affects the activation of the other. Further investigation into how the STAT3 and NF- κ B pathways might influence each other, either through direct molecular interactions or by modulating shared downstream targets, would be valuable to better understand their functional relationship (although this may be for follow up papers). Full sequencing data for the 3 subpopulations in the two genotypes (WT and APP/PS1) are accessible at Gene Expression Omnibus under reference GSE290101. List of DEG between these subpopulations are now provided as **Table 1** (see Reviewer 1, point 6); and two new heatmaps show DEG between the three subpopulations in APP/PS1 and WT mice (**Fig. 5a**).

The double STAT3+/NF- κ B+ subpopulation is systematically analyzed and displayed, except for the proteostatic experiment in **Fig. 5f, g**, because this population overlapped with a population of autofluorescent cells and could not be assessed reliably, as reminded below, in point 10.

We fully agree that the question of cross-talk between the two pathways is very important. Our current data show that depending on the specific gene or function considered, one or the other pathway dominates. For example, the double STAT3+/NF- κ B+ astrocytes have a similar domain size to STAT3+ astrocytes (**Fig. 4b**); but a similar expression profile to NF- κ B+ astrocytes regarding Cts genes (**Fig. 5d**). The molecular dissection of the interaction between these two pathways would require many additional experiments (especially because downstream targets of each pathway can modulate the other). Such experiments are outside the scope of this manuscript, but we have added this point in the discussion.

3) Personally, I found the manuscript difficult to follow with the constant jumps between wildtype and AD conditions. Would it not be more logical to discuss the wildtype situation comprehensively and then move to AD? As it stands, the emphasis of the paper seems focused on AD - is this what is intended? Furthermore, what the authors are really dealing with is a model of Alzheimer's type amyloidosis (APP/PS1) – not Alzheimer's disease *per se*. The manuscript would also benefit from a description of the timeline for plaque deposition and the appearance of behavioral deficits, which was not readily apparent. Finally, if the emphasis really is on the AD/amyloid context, are the effects seen in the APP/PS1 mouse line mirrored in other mouse lines, which would be expected if the results are broadly applicable to AD-type amyloidosis. **Our focus is indeed on the AD/amyloid situation, as we initially aimed at understating how these two cascades linked to astrocyte reactivity are activated in the pathological context. However, as we noticed that amyloid plaques did not significantly impact cascade activity, we also analyzed WT mice to dissect out the intrinsic versus AD-induced effects on astrocyte features. We have thoroughly edited the manuscript to clarify this point and WT data are now shown in main figures for a direct comparison with APP/PS1 conditions. We hope this is now clearer, the manuscript has been revised accordingly.**

We also observed the three subpopulations in equivalent proportion in knockin APP^{NL-F/NL-F} mice, showing that our results may apply to another amyloidosis model. The data are now provided in **Supplemental Fig. 1b**, but we think that characterizing further this model is beyond the scope of the present manuscript.

Finally, we have i) systematically replaced AD mice by APP/PS1 mice, ii) use “amyloidosis” or “AD-related symptoms” instead of AD and iii) included a timeline of some typical AD symptoms in APP/PS1 mice in **Fig. 1a**.

4) I agree with the authors that many of the studies describing astrocyte subtypes have not adequately addressed functionality. In my opinion, whether the authors actually address this issue is, however, debateable. Certainly, I feel that the FRAP measurements of gap junction coupling, hemichannel activity assays and plaque measures provide little extra information than, for example, differential Ca²⁺ signaling in astrocyte subtypes (for example PMIDs: 32139688 and 18575586). **We thank the reviewer for mentioning these 2 publications. Batiuk et al, 2020 performed calcium imaging on astrocytes from hippocampus (CA1), cortical layer I and III-V (PMID 32139688, shown in their Fig. 6). They found differences between Layer I and III-V cortical astrocytes in their mean peak response to norepinephrine, but most differences were between hippocampal and cortical astrocytes. Then, the authors performed hierarchical clustering based on individual calcium responses and showed that astrocytes from these 3 regions may belong to any of the 3 identified functional clusters, without clear segregation (i.e. Layer II-V astrocytes are found in equal proportion in the 3 functional clusters, and thus may display various types of responses). This observation**

supports the idea that within the cortex, several subtypes of functional astrocytes co-exist. However, this type of analysis provides no information on the molecular profile, gene marker or regulatory cascades, for these putative functional astrocyte subtypes.

In the other study mentioned (PMID 18575586), Takata and Hirase compared calcium dynamics in layer 1 and layer 2-3 astrocytes in the rat somatosensory cortex using *in vivo* 2 photon microscopy with a calcium dye. They report some differences, such as calcium transients being synchronous in Layer 2-3 astrocyte processes but not in Layer 1. This is quite different from our study where we report intra-regional functional differences on two very different functions (proteostasis and hemichannel activity), and identify the possible signaling pathways involved. However, we recognize the importance of these pioneer studies in illustrating cortical astrocyte functional heterogeneity and now cite them in the Introduction.

The authors did describe differences in several behavioral tests run following the inhibition of STAT3 or NF- κ B+ signalling. However, these tests were performed largely in the AD/amyloid context and do not provide information (in my mind) on the effects of pathway inhibition in wildtype mice. Hence, I do not think the observed behavioural changes necessarily can be extrapolated to functional importance in the wildtype situation. Finally, did the authors attempt to block both pathways simultaneously and look at the behavioral outcomes (even in the AD/amyloid context)? What were the effects? We performed new behavioral analysis in WT mice and found no effect on anxiety, social preference and memory in these mice (**Supplemental Fig 5e-g**), contrary to APP/PS1 mice (**Fig. 7**), showing that these subpopulations have context-dependent effects on behavior.

We did not block both pathways because we wanted specifically to pinpoint the distinct roles of the two pathways or astrocyte subpopulations. We recognize that it would be interesting to do this experiment, but we think it is outside of the scope of the present manuscript (as mentioned in point 2).

Minor issues:

1) The quality of the figures in the manuscript provided for review was low. Higher-resolution images and clearer figure presentation are needed to ensure data interpretation and reproducibility. We apologize for this. We provided key figures (1, 2, S1) separately from the pdf, at high resolution but maybe the reviewer did not have access to them. All figures are now provided independently, at high resolution. Furthermore, in Figure 2 and Supplemental Figure 1, it would be beneficial if the authors avoided the combination of red and green to ensure figures are accessible to color-blind readers. We are aware of this and we used magenta whenever possible in our panels. In **Fig. 2e** and **Supplemental Fig 1**, we now show the red and green channels separately, to avoid introducing new colors disconnected to the reporter proteins used. Indeed, different reporter proteins (GFP, CFP, Td-Tomato) are used in this manuscript, we wish to keep figure interpretation straightforward by displaying them in their expected color.

2) In general, the statistical analyses used are well described in the 'Methods' section and appear appropriate. Significance is defined as $p < 0.05$. However, in the figures it appears multiple p-values are considered (multiple *), but the significance levels are (generally) not given on the figure itself, or in the legend. Now statistical values linked to */**/***/**** are provided in legend to avoid overloading figures with numbers. P values of interest over 0.05, are displayed on a few panels. Note

that based on Reviewer 3's comments and analysis with linear mixed models when it applies, some *p* values have slightly changed, without changing the conclusions.

3) Full details on the systems and parameters used to acquire fluorescence images should be provided in the 'Methods', rather than merely stating 'Images were acquired with an inverted scanning confocal microscope'. How were thresholds for image analysis set? Some details on image acquisitions were already provided in the next paragraph. We now provide more details and we have gathered all the information under a single section entitled *Image acquisition and analysis*.

As mentioned in the discussion "*Whenever possible, we analyzed absolute GFP/CFP levels, instead of defining binary subpopulations (e.g. Fig. 1f, 1g, 2c, 3b, 3c, 3e, 3f), allowing a precise analysis of STAT3 and NF-κB activity level*". Only when necessary, we applied a threshold to define an analyzed astrocyte as STAT3+, NF-κB+, or STAT3+/NF-κB+. This threshold was defined based on the signal measured in the background and was kept constant for all cells and mice within an analysis. For FACS analysis (RNAseq and proteolysis activity measurement, **Fig. 5, Supplemental Fig. 3**), gates for positive cells were defined relative to non-fluorescent cells.

4) The use of GFAP staining to report morphology has, to the best of my knowledge, largely been discounted in favour of the use of membrane-localized fluorescent reporters, which more accurately report morphology. At the least, the caveats of GFAP use should be discussed. We agree with this limitation but we wanted to avoid using a third viral vector with another fluorescent protein and instead perform a simple assessment of astrocyte domain and cytoskeleton complexity. This simple morphological analysis was indeed sufficient to evidence differences between the three subpopulations. We have added this sentence in the Result section to acknowledge this limitation: *« even if labelling for GFAP+ cytoskeleton does not reveal the full complexity of astrocytes, our simple morphological analysis ...»*

5) In my opinion, the manuscript would be helped by explanations of how the cathepsin and proteasome probes work. We now mention in the corresponding method section: "*These cell permeable activity probes become fluorescent when processed by Cts or the proteasome, respectively* » and in the Result section: "*Cathepsin and proteasome activities were assessed with fluorescent activity probes in single living astrocytes identified by their expression of reporter proteins on a cytometer*".

6) In line 108, only Fig. 1a is referenced, whereas the text description appears to relate to more than just Fig. 1a. This sentence was corrected, in fact the full sentence refers to **Fig. 1a**.

7) In line 493, the figure number associated with the text is missing and should be added. Thanks for pointing this out, this is corrected.

8) In lines 555 and 562, the phrase "data not shown" should be avoided. Please include the relevant data, even as supplementary material, to substantiate the claims made. We have added a histogram showing the density of BAM10+ amyloid plaques as **Supplemental Fig. 5a** and the total distance moved for each behavioral test as **Supplemental Fig. 5b, c, d**.

9) In Fig. 1c, the authors present the quantification of the total percentage of fluorescent cells co-expressing different cell-specific markers; however, they do not provide the values for each marker quantification. From figure, it seems odd that the percentage of astrocytes expressing reporters is so

low (approx 70%) if specificity is high, especially considering the low amount of off-target expression detected in the multiple cell types assayed (< 5%). Why do the combined values not add up to 100%? This rather low percentage was due to GFAP being used as a (reactive) astrocyte marker to assess LV reporter tropism in APP/PS1 mice, as GFAP levels are known to be highly heterogeneous in normal and diseased brains (Escartin et al., *Nat. Neuro*, 2021, PMID: 33589835). As suggested by Reviewer 1, we have now assessed LV reporter tropism by co-immunostaining with the ubiquitous astrocyte marker phosphoglycerate dehydrogenase (PHGDH). We found more than 88% of GFP and/or CFP cells co-expressing PHGDH (Fig. 1b, c). As suggested, we also provide the percentage of PHGDH co-expression by each subpopulation in APP/PS1 mice in Supplemental Fig. 1a. It shows that both reporters have a strong astrocyte tropism and that all three subpopulations are astrocytes. Note that all the co-labelling of Fig. 1c were done on different sections due to antibody incompatibility, explaining that the total may not be equal to 100%.

10) In Fig. 5f and 5g it would be interesting to also show the data regarding the population expressing both reporters. As we mentioned in the Methods of the original version: *“Of note, double CFP+/Td-Tomato+ astrocytes overlapped with autofluorescent cells in the tail of the major negative cell population and could not be analyzed reliably.”* This leads to an over-estimation of the percentage of CFP+/Td-Tomato cells with detectable Cts or proteasome activity and thus we preferred not to include the data.

Reviewer #3 (Remarks to the Author):

Poulot-Becq Giraudon and colleagues describe three molecular clusters of astrocytes based on their NfκB and STAT3 transcription factor activity using novel lentiviral fluorescent reporter probes in WT and APP/PS1 mice co-injected in the mouse prefrontal cortex. They nicely validated the co-injection/expression approach by co-injecting two lentiviral vectors encoding the expression of either GFP or Td-tomato driven by an ubiquitous astrocytic promoter (PGK) followed by FACS and showing a remarkable co-expression in the same astrocytes. Surprisingly, they show that all three subpopulations of astrocytes exist in similar proportions across WT and APP/PS1 mice and that their proportions are not impacted by proximity to amyloid plaques in APP/PS1 mice, suggesting that these three subpopulations of astrocytes are developmentally different by as-yet-unknown epigenetic mechanisms. They show morphological (via Sholl analysis), transcriptomic (via bulk RNA-seq), electrophysiological (via patch-clamp), and hemichannel activity (using ethidium bromide loading) differences between NfκB+ and STAT3+ astrocytes. NfκB+ astrocytes are larger, have more lysosomal activity and perhaps less proteasomal activity, and less hemichannel activity than STAT3+ astrocytes. Inhibition of STAT3 with LV-SOCS3 and of NfκB with LV-NFKBIA reduced plaque size but altered social memory in the 3-chamber test.

Overall, this is a well-written and very interesting paper that not only adds more evidence to the recent literature on the molecular heterogeneity of astrocytes in the normal and AD brain but also provides both innovative tools to study the molecular pathways driving reactive astrogliosis and new insights into the downstream consequences of the activity of NfκB and STAT3, two of the main transcription factors involved in reactive astrogliosis in AD. The characterization of astrocytes in WT and APP/PS1 mice with morphological, electrophysiological, transcriptomics, behavioral, and functional ex-vivo assays is commendable. The notion that these different molecular subpopulations of astrocytes are epigenetically determined in the normal brain is quite intriguing and novel. We thank the 3rd reviewer for the positive evaluation of our work and the insightful suggestions.

I have several constructive questions, comments and suggestions:

1. The choice of prefrontal cortex over hippocampus is not well explained. Certainly, the choice of behavioral tests is adequate to evaluate the function of the prefrontal cortex but is there any particular reason to avoid hippocampus for this approach given its importance in AD? *The PFC was selected because like the hippocampus, it is a vulnerable brain region in AD. But there is also a technical reason in avoiding the hippocampus: the Mokola-pseudotyped lentiviral vectors poorly infect hippocampal astrocytes. The lower transduction efficacy of hippocampal over cortical astrocytes is probably explained by their known molecular differences. We have added this sentence at the beginning of the Result section to clarify this point “The PFC was selected as a vulnerable brain region in AD^{35,36} displaying a high transduction efficiency with Mokola-pseudotyped LV”.*

2. I am a bit surprised that proximity to plaques did not influence reporter expression. Have prior studies shown p65 and pSTAT3 immunoreactivity in reactive astrocytes around plaques? The fact that SOCS3 and NFKBIA lentiviral expression reduced plaque size also argues that STAT3 and NfκB are activated in astrocytes around plaques. I have several suggestions to reanalyze and/or clarify these analyses:

a. It may be unfair to include NfκB+ astrocytes in the GFP fluorescence vs. distance to plaque regression in Fig 3b and STAT3+ astrocytes in the CFPnuc fluorescence vs. distance to plaques in Fig. 3c since they were classified as NfκB+ or STAT3+ based on their low fluorescence intensity for GFP and CFPnuc, respectively. If the regression is repeated excluding the “negative” group there may be a statistically significant negative correlation between GFP or CFPnuc fluorescence and distance to the nearest plaque. *Thank you for this suggestion, the regression analysis was done with or without the negative subpopulation and it does not change the result (now mentioned in the manuscript).*

b. If the denominator of the proportions in Fig. 3d is the total number of astrocytes of each cluster, isn't the fact that the proportion of the three clusters of astrocytes is highest in the 41-80 microns interval evidence that there is an effect of plaque proximity? *We apologize if this figure was not clear. Instead, this initial figure should have been understood as showing that the majority of astrocytes (40+20%) are less than 80 μm from a plaque, whatever their subpopulation. We now provide a more straightforward histogram in Fig. 3d, showing that for each bin of distance to a plaque, the three astrocyte subpopulations are observed in the same proportions. This illustrates that plaques do not induce a specific subpopulation around them.*

c. The distance to the nearest plaque was measured as the distance between the center of the plaque and the center of the astrocyte cell body. Therefore, this distance includes the plaque diameter, which can be anywhere from 5 to 100 microns depending on the size of the plaque. Astrocyte cell bodies are not embedded in the amyloid plaque (unlike microglia cell bodies, which may be) but they are pushed away by the plaque mass effect (see repulsion effect described by Galea E et al. PNAS 2015 in the same mouse model). This explains why there are few astrocytes in the first bins and why the proportion for all three clusters is lower in the 0-40 microns bin. Measuring distance from the edge of the nearest plaque would have prevented this issue and likely shown an effect of plaque proximity. *We thank the reviewer for this very good point. We have measured the distance between astrocytes and the edge of the closest plaque, and it does not change the results. Fig. 3b-f were updated with the new data.*

d. Can these results be validated with Abeta, p65 and pSTAT3 multiplex immunohistochemistry? If the conclusion is that plaque proximity did not make a difference in reporter fluorescence, could the reason be that reactive astrocytes around amyloid plaques were less amenable to lentiviral transduction due to an antiviral interferon-mediated response to plaques as shown by the Liddelow lab? Please, comment on this possibility. It is a good point, but we did observe infected astrocytes in close proximity to plaques (see image in Fig. 3a, and quantification in Fig. 3b, c). Now that we display the distance between fluorescent astrocytes to the edge of the plaques, it is clear that astrocytes can be infected and fluorescent even within a 20 μm distance from plaques, ruling out this possibility. This is now mentioned in the text. We provide a different image in Fig. 3a showing examples of astrocytes of the three subpopulations around amyloid plaques, with improved annotation to illustrate this point.

Also note that the detection of phosphorylated epitopes on p65/NF-kB and STAT3 is really difficult on AD mouse models, probably because they are easily degraded postmortem or are present at low levels in these chronic models. This is why we chose to develop these LV reporters in the first place, to have an alternative and more sensitive strategy to detect pathway activity *in situ* (see reply to reviewer 1, point 1, now mentioned in the manuscript). Indeed, we detect many more GFP+ or CFP+ astrocytes (i.e. with active pathway), in the mouse brain, even away from plaques than the few pSTAT3 astrocytes around large plaques reported by Reichenbach et al 2019 for example.

3. Were the morphological and ephys analyses of the three astrocyte subpopulations controlled for cortical layer? In other words, could the differences noted across molecular subtypes be due to differences in the layer location when selecting the astrocytes to be analyzed? While the authors did not find differences in subtype proportions across layers, a layer-by-layer comparison of these morphological parameters could be helpful to ensure that the cortical layer where the astrocyte sits is not driving the differences observed. As we observed no subpopulation enrichment in a specific cortical layer but instead a random distribution along the needle track (see Fig. 2b), we did not take the cortical layer into account in our analysis. Astrocytes from all PFC layers are expected to contribute homogeneously to measurements in the three subpopulations. Of note, electrophysiological recording was always performed in deeper layers (layers V-VI). We mention this in the corresponding method section: *“To limit potential differences between cortical layers, astrocytes were consistently recorded in deep cortical layers (layer V-VI), and astrocytes belonging to the three subpopulations were recorded along the same layer, within each slice, for each mouse.”*

4. RNA-seq data:

a. Were pathway enrichment analyses in Table 1 done with a rank-based gene set enrichment analysis or using a hypergeometric overrepresentation test? Is the p-value in Table 1 adjusted for multiple hypothesis testing (e.g. FDR)? Should the pathway enrichment analyses in Table 1 be done separately for the 477 up and the 158 down DEGs in STAT3+ vs. Nfkb+ astrocytes? What are the pathways enriched in Nfkb+ vs. STAT3+ astrocytes in both APP/PS1 and WT mice? Also, the KEGG database is very biased towards infectious pathogens. Could the authors try GO-BP and Reactome databases for their pathway enrichment analyses? We thank the reviewer for the insightful comments. As the initial Table 1 displayed p values only for KEGG analysis, we have extended this analysis according to these suggestions. The enrichment analysis was performed only on up or down-regulated genes, using Reactome, KEGG, and GO-BP with FDR provided as statistics (overrepresentation tests; new Table 3). This extensive analysis still highlights differences in proteostatic pathways (new Fig. 5c) but also reveal a more inflammatory profile of the NF-KB+

subpopulation in APP/PS1 mice, which is now discussed in the Results section. Analysis of the WT subpopulations and Methods were also further developed.

b. Transcription factors upstream DEGs were evaluated using Pscan, which is based on the presence of known binding motif sequences, however this sequence-based approach could yield non-specific transcription factors. A Transcription Factor Enrichment Analysis based on the ENCODE ChIP-Seq database would be helpful to complement this analysis. We performed a Chip-seq data analysis using ChEA3-ENCODE. The results are consistent with Pscan analysis, they are further discussed in the main text and are now provided in Table 2.

c. Fig. 5c: Do astrocytes express Ig Fc receptors, Clec7a, Cd68? Aren't these microglial genes? Could the authors provide a supplementary Excel spreadsheet with the differential expression analysis results at the gene level? There are several reports of astrocytes expressing typical gene markers of other cell types at low levels, especially in the mouse cortex. For example, Morel et al. *Glia*, 2018 (DOI: 10.1002/glia.23545) describe an astrocyte subpopulation (EAAT-Td-Tomato^{low}/Aldh1L1-eGFP+) that express significant levels of *Iba1*, *CD11b*, *Mog*, *Mbp* and *pdgfra*, markers of microglia, oligodendrocytes and OPC respectively. Likewise, John-Lin et al., *Nat. Neuro.* 2017 (DOI:10.1038/nn.4493) report a cortical astrocyte subpopulation positive both for Aldh1l1 (Aldh1l1-GFP+) and CD63, another microglial gene. This can be further exacerbated upon pathological conditions, as reported by Wilhelmsson et al, *Cereb. Cortex*, 2017 (DOI: 10.1093/cercor/bhx069), who observed cortical cells with an astrocyte morphology, co-expressing GFAP, Aldh1L1 and Tmem119 (a microglia marker) or S100b and IBA1 in patients with stroke, Lewi body dementia or AD. These cells expressing dual markers were more frequent in pathological conditions. Likewise, O'Shea et al. *Nat. neuro*, 2024 show that typical microglial genes are up-regulated in reactive astrocytes following spinal cord injury (DOI: 10.1038/s41593-024-01684-6). This is in accordance with our own data, where the typical microglial markers (*Fcer1g*, *CD68*, *Clec7a*) are significantly enriched in NF-kB+ compared to STAT3+ astrocytes, only in the pathological AD context. In fact, other highly expressed microglial genes such as *Csf1r*, *Tmem119*, *Trem2* or *Cx3cr1* were not among the DEG, suggesting that the NF-kB+ subpopulation acquires a specific inflammatory phenotype more similar to microglia, in a disease context. This is now discussed in the manuscript and **Table 1** is provided with full lists of DEG for all comparisons performed.

5. Can the authors speculate why SOCS3 expression only lowered GFP fluorescence by 26% after 3 months? Could this be due to longer half-life of GFP vs. CFP? Why the authors think this is enough to trigger changes in plaque size and behavior?

SOCS3 partial inhibitory effects on STAT3 reporter is not unexpected as STAT3 can be activated by other upstream kinases than JAKs (e.g. MAP kinases, tyrosine kinase-coupled receptors). Even in presence of SOCS3, there could be a remaining STAT3-dependent transcriptional activity through these non-canonical pathways or even by unphosphorylated STAT3, which was shown to bind DNA and regulate transcription (see refs. 16, 70, 71 in the manuscript). Finally SOCS3 was expressed under the control of the P-STAT3 promoter which is weaker than the P-PGK promoter used in our previous studies (refs. 18, 19, 33), probably explaining this partial effect.

Still, SOCS3 inhibitory effect on the STAT3 pathway was enough to impact plaque size and behavior, although modestly. We can speculate that a stronger or spatially extended pathway inhibition would have larger effects. This is now discussed.

6. Statistics: were astrocytes from the same mouse treated as independent observations in one-way ANOVA and 2-way ANOVA or did the authors use the average per mouse in the ANOVA? The former would imply pseudoreplication and mixed effect models are recommended to avoid it. We apologize

if this was not clear, and we are well aware of the pseudo-replication issue. For all analysis comparing metrics between the three subpopulations, we initially used a 2-way ANOVA with GraphPad Prism, where each analyzed astrocyte belongs to a given subpopulation and a mouse, to take into account the “mouse effect”. The statistics provided were the p-value for the “subpopulation” effect. However, we are unsure whether Prism treats the mouse factor as random. Hence, when appropriate, we re-analyzed our data with R using a linear mixed model with “subpopulation” (and sometimes “genotype”) as fixed factors and “mouse” as a random factor, following this very insightful suggestion. This analysis does not change our conclusions and even provides more significant p values in some plots. The exact statistics are mentioned in corresponding panel legends. For clarity, our plots show the average value per mouse but the number of individual astrocytes analyzed is provided in legends.

Minor:

1. Lines 186-187. The authors state “The PFC of three mice were pooled, resulting in N=5 independent PFC samples from APP mice.” However, Figure 2d shows: Proportion of each astrocyte subpopulation, obtained from three independent cohorts with each N=6-7 APP mice. Can the authors reconcile these apparently discrepant statements? If the cell sorting experiment was repeated 3 times, can they show an additional stack bar plot with average + s.d. or s.e.m. error bars in Figure 2d? We apologize if this was not clear: Mice were pooled only for RNAseq analysis to sort enough astrocytes for proper sequencing. **Fig 2** shows proportion calculated from dot-plots generated from individual cell analysis of histological sections (as mentioned lines 166-167) not FACS. To have a better estimate and because we included independent mouse cohorts for the different analyses provided, we combined histological data obtained from four independent cohorts. To clarify this, we have added “on histological PFC sections” in **Fig 2** legend.
2. Throughout the manuscript, please consider referring to APP mice as APP/PS1 mice. This is changed throughout the manuscript (except for **Fig. 7** and **Supplemental Fig. 5**, for clarity).
3. Suppl. Fig. 3: I believe the authors mean singlets rather than singulets.
4. Suppl. Fig. 4b: Please, consider labeling panel as “photobleach” rather than “bleach.” Thank you for pointing out these two mistakes, both are corrected on the manuscript and figures.

REPLY TO REVIEWERS

We thank all 4 reviewers for their careful reading of the revised manuscript. We are delighted that they acknowledge its improvement. We are grateful for their positive assessment, and minor comments to further improve the manuscript.

We have addressed all points below.

REVIEWER COMMENTS

Reviewer #1

The authors have responded to all of the points raised in my first review with new experiments, rearranging of existing data, and by adding clarifying statements to the text. I have no further concerns.

Three minor points:

1. Given Reviewer 3's query about 'microglia' genes being detected in the sequencing data, a suggestion is to also include expression level data in the supplementary RNAseq table to give an indication of how highly these genes are expressed in the astrocyte samples. We agree that it would be interesting to show the level of expression of typical astrocyte and other cell-type genes. However, these data only make sense when compared with their level in their respective cell types (i.e. microglia genes can be detected in astrocytes but at lower levels than in microglia). As we noted in the Material & Methods section, we did not analyze non fluorescent cells because these are composed by a mix of different cell types including non-fluorescent astrocytes. Therefore, the specific enrichment or de-enrichment for astrocyte or other cell type markers cannot be quantified in our sorted subpopulations. We now explain this further in Material & Methods: *“Non-fluorescent cells were not analyzed because they were composed of very heterogeneous cells (microglial cells, neurons, oligodendrocyte progenitor cells but also many non-transduced, non-fluorescent astrocytes), and thus it was not possible to compute an enrichment score for astrocyte markers over other cell type markers.”* However, we are confident in the specificity of our astrocyte sorting as LV reporter tropism was validated in **Fig. 1b-c**, and many typical astrocyte genes were consistently detected in sorted astrocytes (e.g. *Apoe, Glul, AldoC, connexins* see **Fig. 6a**).

In the Discussion, we also edited the section on microglial genes as: *“Of note, NF-kB+ cells were found to express several microglial genes at higher levels than STAT3+ and STAT3+/NF-kB+ astrocytes in APP/PS1 mice. Detection of genes used as microglia markers in sorted astrocytes is not unexpected, as there are several reports of astrocytes expressing microglia-enriched genes in various pathological conditions ^{40,41} and even in physiological conditions ^{64,65}. The fact that other highly expressed microglial genes such as *Csf1r, Tmem119, Trem2* or *Cx3cr1* were not among the DEG suggests that NF-kB+ astrocytes acquire a specific inflammatory phenotype in APP/PS1 mice.”*

2. Figure 5D, STAT3/NFKB column for the APP sample - there is an extra datapoint in the column (3 instead of 2) Thanks for pointing this out. This is corrected.

3. The GEO record is set to private so could not be accessed. A GEO token for reviewers was provided in the reporting summary. We are sorry if the reviewer missed it. The GEO record will be made public when our manuscript is accepted.

Reviewer #2

In this study, Giraudon and colleagues use astrocyte-specific reporters for the STAT3 and NF- κ B signalling pathways to identify at least three astrocyte subpopulations in the prefrontal cortex (PFC) of an Alzheimer's disease (AD) mouse model. Notably, these signaling-defined subpopulations are not induced by amyloid deposition and are also present in wild-type mice. These subpopulations exhibit distinct morphologies, molecular signatures, and functional profiles. NF- κ B⁺ astrocytes occupy larger territories and show increased lysosomal activity, whereas STAT3⁺ astrocytes display enhanced connexin hemichannel opening. Selective inhibition of these subpopulations in AD mice reduces amyloid plaque size in the prefrontal cortex and modulates behaviours such as anxiety, social preference, and memory. Hence, the study demonstrates that intrinsic signaling activity defines astrocyte subpopulations in the mouse PFC and highlights that these subpopulations exert distinct and complex effects on hallmark AD alterations.

This is a resubmission of a previously reviewed manuscript.

Overall, this is a well written and interesting paper that adds considerable weight to our understanding of astrocyte (molecular) heterogeneity in both the healthy and diseased brain. It is obvious the authors made substantial efforts during the revision and, as such, the manuscript is very much improved. However, I would like clarification regarding experimental data added to the manuscript and raise a number of minor comments for consideration by the authors.

Major Issue:

i) The authors added new behavioral data on wild-type animals which shows no effects on anxiety, social preference and memory in these mice following pathway inhibition. This is in contrast to the effects seen in APP/PS1 mice. From the manuscript, it appears that the wild-type testing was done during the revision period (independent cohort). However, I am not sure that the data obtained and comparisons made are as reliable as implied (e.g. Fig 7c-f, Supp Fig 5 a-d) - as sensitivity to the different testing conditions that were undoubtedly present may impact animal performance in an unpredictable way. *Indeed, we did not include the WT-SOCS3 and WT-NFKBIA groups in the original behavioral study. It would have resulted in a large number of mice (N=70), difficult to handle simultaneously and without introducing experimental biases in lengthy tests like the three-chamber test. The additional data shown in Supplemental Fig. 5e-g with WT mice were generated secondarily during revisions, but in the same facility, using the same devices and performed by the same experimenter, following standardized protocols. This new cohort contains the appropriate internal control group (WT-CTR), which behaved as expected, with performances on the EPM and three-chamber tests similar to the equivalent group in the first cohort (see Fig. 7c-f, and Fig. S5e-g). We agree with the reviewer that groups from the two cohorts cannot be statistically compared because they were done independently. This is why we did not directly assess differences between WT-SOCS3 and APP-SOCS3 groups for example. Instead, we interpreted SOCS3 effects within an APP or WT genotype, by comparing APP-SOCS3 to APP-CTR or WT-SOCS3 to WT-CTR respectively. This is now clarified in the Result section: "Even if groups from the two independent cohorts cannot be directly compared, these results suggest that SOCS3 and NFKBIA influence mouse behavior differently in healthy and disease conditions."*

Minor Issues:

i) In my opinion, there are instances of hyperbole which can (and should be) toned down, without the paper losing any of its impact. For example, the authors claim 'results show that PFC astrocytes display signaling heterogeneity in WT mice and throughout disease progression in APP/PS1 mice' when the latest timepoint they assess in the mice is 12 months of age. APP/PS1 mice have a much longer lifespan (Roberts et al. doi: 10.1101/2024.10.15.618508). *This is a fair point and we have changed the sentence to: "These results show that PFC astrocytes display signaling heterogeneity in WT and APP/PS1 mice."*

Likewise, I am still of the opinion that the authors provide no direct proof of functional differences in the text (original review point 4). What they actually report is differences in proteasome activity using reporter assays, SR101 diffusion between gap junctions and ethidium bromide uptake via hemichannels. While this is highly suggestive of functional alterations, there is no direct evidence that the functional output of the cell is changing. Likewise, while the authors show an effect of astrocyte manipulation on behavior, it is unclear exactly what the underlying mechanistic action driving this effect is. The papers cited in review (Batiuk et al. and Takata and Hirase) reported distinct physiologies (Ca²⁺ dynamics) of cells without a clear functional output. *We understand the reviewers' point. In this paper, the functional characterization of astrocytes relates to their **intrinsic functions** but it is true that we did not assess the output effects on neurons or other cell types. This is much harder to achieve, given that the different astrocyte subpopulations are intermingled in the PFC. We now acknowledge this point in the Discussion by using "intrinsic functions" and saying "The functional impact of the three astrocyte subpopulations on neuronal circuits is however quite challenging to test experimentally, given the lack of methods to probe gliotransmitter release by single cells."*

ii) On reflection, I agree with Reviewer 1's comment on the number of samples used for RNA-seq. I do not necessarily disagree with the author's response, especially given the data they show in the rebuttal, but I do think that some reference to the low sample number (and the limits of interpretation) should be made. *This is also a fair point and we acknowledge this limitation at the beginning of the Result "Despite low number of replicates due to limited number of sorted astrocytes and stringent quality criteria applied (see Methods), we identified hundreds of differentially expressed genes (DEG) with high fold-change between these three subpopulations in APP/PS1 mice."*

iii) I found it hard to interpret the DEG numbers in Fig. 5A and B: can the authors please clarify? *We have edited panels 5A and 5B, and clarified the legend.*

iv) The inhibition of the NF-κB pathway in wild-type mice has an effect on the total distance covered in the 3-chamber test, but, as far as I can see, any discussion of this is omitted from the manuscript. Why? How should it be interpreted? *WT-NFKBIA mice did not display general hypolocomotion, as they covered similar total distance to WT-CTR mice at the EPM. Their lower total distance travelled specifically at the three-chamber test, with yet a preserved interest for both stimuli, is hard to explain. This result was clearly described at the end of the results "Mice in the WT-NFKBIA group showed a reduction in total distance moved during the three phases of the three-chamber test (but not at the elevated plus maze), suggesting reduced mobility in this test, despite similar interest in the different stimuli presented (Supplemental Fig. 5g)". We think that further discussing this observation would be too speculative and divert the reader from the main focus of the article.*

Reviewer #3

The authors have satisfactorily answered all my questions and comments. I appreciate the additional bioinformatic analyses, the reanalysis of distance to plaque, and the application of mixed effect models. As per requests from the other reviewers, the authors have added in vitro validation experiments in HEK cells and replicated the three subpopulations of astrocytes (STAT3+, NF-κB+ and STAT3+/NF-κB+) in the APP-NL-F/NL-F mouse model.

Although I did not ask for the in vitro validation experiments of the probes, they seem to confirm a point I raised, that the STAT3-GFP probe seems to be less responsive to known STAT3 inhibitors (Stattic, SOCS3) than the NF- κ B probe to known NF- κ B inhibitors (BAY, NFKBIA) as judged by the differences in effect size in Figures 1d vs. 1e and 1f vs. 1g. Could the authors comment whether these differences are more likely to be technical or biological? What are the implications of such difference for the interpretation of the work? *This is a good point. We now acknowledge this in the Discussion: “We noted that the STAT3 reporter was less sensitive to pathway inhibition than the NF- κ B reporter. This may be related to a lower efficacy of the pharmacological and genetic inhibitors used. Different mechanisms of action are at stake: NFKBIA directly binds NF- κ B and sequesters it in the cytoplasm while SOCS3 acts upstream on the STAT3 pathway, by inhibiting receptor-bound-JAK⁵⁰. The STAT3 reporter could still be activated by non-canonical STAT3 pathways, independent of JAK-mediated STAT3 phosphorylation^{16,51,52}.”*

We already discussed that the partial effects observed with inhibitors in **Fig. 7** would probably be enhanced with a stronger and spatially extended SOCS3 expression: *“These effects were rather mild, which can be expected given that only a small fraction of the PFC was targeted with this LV strategy and the expression of the genetic inhibitor was not driven by a strong ubiquitous promoter. We can thus speculate that a stronger or spatially extended pathway inhibition would have larger effects.”*

Why not using primary astrocytes or an astrogloma cell line? *We did try murine primary astrocyte cultures but obtained very low transduction rates with our LV reporters pseudotyped with the Mokola envelope, precluding a reliable analysis with pharmacological inhibitors.*

Minor points:

1. For the readership, it'd be great if the overlapping genes in Table 3 are expressed as gene symbols in addition to Ensembl ids. *Thanks for the suggestion, Table 3 was edited accordingly.*
2. Introduction and Material and Methods - Animals: please add the Greek letter beta before amyloidosis. *This was corrected throughout the manuscript, including in **Fig. 1a**.*
3. What do the white and black circles mean in new Figures 1d-e, cultures #1 and #2? Please, explain in the legend. *Thank you for pointing this out we have clarified “**d, e, black and white circles represent the two culture replicates.** »*